



# Assimilation of vegetation optical depth retrievals from passive microwave radiometry

Sujay V. Kumar[1], Thomas R. Holmes[1], Rajat Bindlish[1], Richard de Jeu[2], and Christa Peters-Lidard[3]

[1]Hydrological Sciences Laboratory, NASA Goddard Space Flight Center, Greenbelt, MD, USA
[2]Vandersat
[3]Hydrosphere, Biosphere, and Geophysics, Earth Sciences Division at NASA GSFC, Greenbelt, MD

*Correspondence to:* Sujay V. Kumar
(Sujay.V.Kumar@nasa.gov)

**Abstract.** Vegetation optical depth (VOD) retrievals from passive microwave sensors provide estimates of above-ground canopy biomass. This study presents the development and analysis of assimilating VOD retrievals from X-, C-, and L-band passive microwave instruments within the Noah-MP land surface model, over the Continental U.S. The results from this study demonstrate that the assim-

ilation of VOD retrievals have a significant beneficial impact on the simulation of evapotranspiration and GPP, particularly over the agricultural areas of the U.S. The improvements in the water and carbon fluxes from the assimilation of VOD from X- and C-band sensors are found to be comparable to those obtained from the assimilation of vegetation indices from optical sensors. The study also quantifies the relative and joint impact of assimilating surface soil moisture and VOD from SMAP.

The utility of soil moisture assimilation for improving ET is more significant over water limited regions, whereas VOD DA is more impactful over energy limited domains. The results also indicate that the independent information on moisture and vegetation states from SMAP can be simultaneously exploited through the joint assimilation of surface soil moisture and VOD. Since the passive microwave-based VOD retrievals are available in nearly all weather conditions, their use within data

assimilation systems offers the ability to extend and improve the utility obtained from the use of optical/infrared based vegetation retrievals.

## 1 Introduction

Remote sensing estimates of vegetation are typically developed by exploiting the relationship between the stomatal stress and the spectral reflectance of leaves and canopies (Knipling (1970)).

Multi- and hyperspectral optical and thermal satellite sensors have been used to provide retrievals of


variables such as Leaf Area Index (LAI), Normalized Difference Vegetation Index (NDVI), fraction of photosynthetically active radiation (fPAR), solar induced fluorescence (SIF), and biomass (Myneni et al. (2002); Tucker et al. (2005); Zheng and Moskal (2009); Myneni et al. (2011); Kumar and Mutanga (2017)). The multi-spectral vegetation indices are typically derived from atmospherically

corrected bidirectional surface reflectance in the near-infrared and visible bands (Price and Bausch (1995); Huete et al. (1997)). Similarly, hyperspectral imaging is used to characterize vegetation type, health and function (Goetz et al. (1985)) at very fine ($\sim$30 m) spatial resolution. As vegetation stress and stomatal closure influence canopy temperatures, thermal remote sensing also offers the possibility of estimating vegetation conditions. For example, Landsat (Anderson et al. (2012))

and the ECOsystem Spaceborne Thermal Radiometer Experiment on Space Station (ECOSTRESS; ecostress.jpl.nasa.gov) provide high resolution ($\sim$ 70 -100 m) estimates of surface temperature and evapotranspiration. Houborg et al. (2015) presents a summary of the major advances in the remote sensing of vegetation from these platforms.

A significant shortcoming of the optical/thermal infrared (TIR) sensors is that cloud cover can

severely limit the acquisition of data, restricting the coverage to cloud-free, clear days. Gap-filling strategies, such as using the nearest clear-day observation, are often used to improve the spatio-temporal coverage from optical/TIR instruments (Hall et al. (2010)). Passive microwave measurements, on the other hand, are nearly all-weather and are not limited by cloud cover. Holmes et al. (2016), for example, used microwave estimates of land surface temperature as an alternate to TIR

measurements to retrieve evapotranspiration (ET) during cloudy time periods. Microwave radiometry over land has traditionally been used for retrieving estimates of surface soil moisture, by exploiting the sensitivity to low frequency microwave radiometric measurements to changes in soil moisture (Njoku and Entekhabi (1996)). As this radiance passes through vegetation, the microwave signal is attenuated by vegetation, the level of which is described by the vegetation optical depth

(VOD) parameter. Due to its sensitivity to plant water content, VOD can be used as an analog of above-ground canopy biomass (Owe et al. (2001); Liu et al. (2011b); Konings et al. (2016, 2019)). The VOD retrievals from various microwave frequencies such as K-, X-, C- and L-bands have been used for a variety of studies for examining vegetation seasonality (Jones et al. (2012)), characterization of extremes such as drought (Liu et al. (2015); Konings and Gentine (2017); Smith et al.

(2020)), assessment of dryland vegetation dynamics (Andela et al. (2013)), and the determination of land degradation and deforestation (Liu et al. (2013); van Marle et al. (2016)).

Despite the availability of vegetation measurements from various sensing platforms, the incorporation of these measurements within data assimilation systems for land surface hydrology is relatively new. Most studies to date have focused on the assimilation of LAI retrievals to improve the

characterization of vegetation biomass, evapotranspiration, root zone soil moisture and $CO_2$ fluxes within land surface models (Sabater et al. (2008); Barbu et al. (2011, 2014); Albergel et al. (2017, 2018); Fox et al. (2018)). More recently, Kumar et al. (2019b) demonstrated the beneficial impact of



LAI assimilation on improving water, energy, and carbon fluxes over the Continental U.S. (CONUS). Most prominent improvements from LAI assimilation are observed over the agricultural areas, where
assimilation improved the representation of vegetation seasonality impacted by cropping schedules.

As the use of all-weather VOD measurements from microwave sensors provides the opportunity to extend the spatial and temporal coverage of vegetation observations, here we examine the influence of assimilating VOD retrievals from microwave radiometry. Specifically, we explore the utility of assimilating VOD retrievals from X-, C-, and L-band microwave sensors in the Noah multi-
parameterization (Noah-MP) land surface model (LSM). The study uses VOD retrievals from a range of microwave frequencies, as their current and future availability vary significantly. For example, the L-band sensing platforms such as SMAP are relatively new whereas the X-band and C-band retrievals of VOD are available for significantly longer time records, with observations from multiple satellites. In addition, given the plans for sensors operating in the higher microwave frequencies (e.g.
AMSR, GMI, JPSS-2), future observations in X- and C-band frequencies are also likely guranteed. Quantifying the relative utility of VOD retrievals in these frequencies is, therefore, important. The model simulations are conducted over CONUS in the North American Land Data Assimilation System phase-2 (NLDAS-2; Xia et al. (2012)) configuration. As noted in prior data assimilation studies such as Kumar et al. (2019b), NLDAS-2 configuration provides an environment with high-quality
boundary conditions informed by radar and gauge corrected precipitation and bias-corrected short-wave radiation, which also leads to high skill in the simulated land surface conditions. Partly as a result of the high skill of the NLDAS-2 meteorology, data assimilation of variables such as soil moisture and snow has only reported marginal success in this configuration (Kumar et al. (2014)). The assimilation of LAI, on the other hand, has been more impactful as it was shown to detect im-
pacts of agricultural activity, which is not easily captured through high quality boundary conditions alone (Kumar et al. (2019b)). In general, demonstration of additional improvements through data assimilation in the NLDAS-2 configuration is indicative of the significant utility of remote sensing inputs, as such high quality boundary conditions are not routinely available in other regions of the world.

As described in detail in Konings et al. (2017), a number of approaches have been used to retrieve VOD from microwave sensors. Here we employ VOD retrievals primarily from two approaches for data assimilation. The Land Parameter Retrieval Model (LPRM; Owe et al. (2008)) uses single frequency, polarized brightness temperature in the range of 1-20 GHz to retrieve both soil moisture and VOD. Here we make use of the C-band (6.9 GHz) and X-band (10.7 GHz) frequency channels of the
Advanced Microwave Scanning Radiometer - Earth observing system (AMSR-E) aboard NASA's Aqua satellite and the AMSR2 instrument onboard the Global Change Observation Mission-Water (GCOM-W). The LPRM approach is used to jointly derive VOD and soil moisture from dual polarized measurements (Owe et al. (2008)). The C- and X-band measurements are less sensitive to cloud water content and more sensitive to soil moisture and vegetation canopy. AMSR observations



at C- and X-band frequencies are also prone to Radio Frequency Interference (RFI). NASA's Soil
Moisture Active Passive (SMAP; Entekhabi et al. (2010)) mission operates in a protected L-band,
which minimizes the impact of RFI contamination. The sensitivity of L-band to cloud water content
is lower compared to C- and X-band. In addition, the L-band measurements provide more sensitivity
to deeper soil moisture and canopy layers.

To our knowledge, this is one of the first reported studies of continental scale assimilation of VOD
retrievals within LSMs. Specifically, this article addresses the following research questions:

  – What is the impact of assimilating VOD retrievals from X-, C-, and L-band passive microwave
    remote sensing instruments on water and carbon states?

  – How does the utility of passive microwave VOD assimilation compare to that of assimilating
vegetation (LAI) retrievals from optical instruments?

  – Does assimilating L-band VOD provide independent benefits to that from incorporating sur-
    face soil moisture retrievals? Can improved simulation of water and carbon states be developed
    from the simultaneous use of VOD and soil moisture?

These questions are addressed by examining the impact of assimilation with the use of a large
suite of independent reference datasets. Section 2 describes the details of the model configuration,
datasets used, and the assimilation configuration. The results of various data assimilation simulations
are described in 3. Finally Section 4 summarizes the main findings of the study.

## 2 Study settings

### 2.1 Data

VOD, an integrated measure of the vegetation structure and water content, is typically estimated
as part of the radiometric soil moisture retrieval approach based on the first order $\tau$-$\omega$ model (Mo
et al. (1982)). In this model, the L-band brightness temperature ($T_{b,p}$) estimates at the top of the
atmosphere for horizontal and vertical polarizations (denoted by the subscript $p$) is represented as:

$$T_{b,p} = T_s(1 - r_p)\gamma + T_c(1 - \omega)(1 - \gamma)(1 + r_p\gamma) \tag{1}$$

where $T_s$ is the surface soil temperature, $T_c$ is the canopy temperature, $r_p$ is the rough surface
reflectivity, $\omega_p$ is the scattering albedo, and $\gamma$ is the vegetation transmissivity. $\gamma$, which represents the
attenuation of the emission due to vegetation is a function of VOD and the measurement incidence
angle $\theta$.

$$\gamma = \exp-(\frac{\text{VOD}}{cos\theta}) \tag{2}$$



VOD is determined by the canopy structure and the dielectric properties of the canopy layer. When VOD is low ($\sim 0$), the attenuation of the microwave signal is small. Soil moisture is estimated from $r_p$ using Fresnel equations that relate $r_p$ to the dielectric constant of the soil. A more detailed description of the VOD formulation is provided in Grant et al. (2016).

As noted earlier, the X- and C-band based VOD datasets used in this study are based on LPRM to
retrieve VOD and soil moisture from dual-polarized passive microwave observations. LPRM uses $\tau$-$\omega$ model to characterize the emission and radiative transfer of low-frequency (1-20 GHz) microwave emission from the soil, vegetation, and atmosphere to the top-of-atmosphere brightness temperature recorded by the satellite. Unique to LPRM, the method includes the analytical solution of the tau-omega model for polarized emission that describes the relationship between the microwave po-
larization difference ratio (MPDI) and VOD (Meesters et al. (2005)). Within the framework of the $\tau$-$\omega$ model, this allows for the retrieval of both VOD and soil moisture and has been implemented with all existing passive microwave satellites with frequencies from L- to Ku-band and from 1979 to present (Owe et al. (2008); Parinussa et al. (2011); der Schalie et al. (2016)). The spatial resolution of this product is 0.25 degree with a global extent of the non-frozen land surface. The temporal
resolution is 1-2 days for the morning overpass. In this study, we employ the VOD retrievals from LPRM version 6 (Van der Schalie et al. (2018)), available from the VOD climate archive (VODCA; Moesinger et al. (2019)).

The SMAP satellite launched in January 2015 is a mission dedicated to measuring soil moisture and freeze/thaw states, employing a L-band passive microwave radiometer. The VOD retrievals from
SMAP are also developed using the $\tau$-$\omega$ model. Though the sampling resolution of the SMAP radiometer is approximately 36 km, oversampling of the antenna overpasses is used to enhance the spatial resolution to 9 km. This 9km, level 2 SMAP dataset (SPL2SMP_E) is used in this study.

## 2.2 Model configuration

The model domain used in this study covers the Continental United States (CONUS) with an extent
of 25-53°N and 125-67°W at 1/8° spatial resolution (Figure 1). Hourly NLDAS-2 meteorological inputs are used to drive the Noah-MP land surface model (version 3.6), which is the next generation version of the Noah LSM. Compared to Noah, Noah-MP provides multiple options for various land surface physics computations, including multilayer snow pack, options for surface water infiltration, runoff, and groundwater, representation of an unconfined groundwater aquifer, and a dy-
namic vegetation model (Niu et al. (2011); Yang et al. (2011)). Note that the prognostic vegetation model of Noah-MP v3.6 was used by Kumar et al. (2019b) to demonstrate the impact of assimilating LAI retrievals from the Moderate Resolution Imaging Spectroradiometer (MODIS) aboard the Terra and Aqua satellites. In addition to Noah-MP, the Hydrological Modeling and Analysis Platform (HyMAP; Getirana et al. (2012)) model is used to develop estimates of routed streamflow using



the gridded surface runoff and baseflow fields from Noah-MP. In this study, the impacts of regulation
and reservoir operations on streamflow are not modeled within HyMAP.

The model and data assimilation integrations in this study are conducted during a time period of
2000 to 2018. The initial conditions are generated through a long spinup of Noah-MP. The model is
initialized with uniform conditions and is run from 1979 to 2018 twice. It is then reinitialized in 1979
with the climatological average conditions derived from the spinup. Finally, the initial conditions at
the beginning of year 2000 are used for the model simulations in this article.

The NASA Land Information System (LIS; Kumar et al. (2006)) is used to facilitate the model
simulations presented in this article. LIS is a comprehensive land surface modeling system that in-
cludes the interoperable support for a large suite of land surface models, data assimilation algorithms,
and observational data sources. As part of this study, the DA capabilities in LIS is extended to en-
able the assimilation of VOD retrievals, described in Section 2.3. The LIS framework also includes
a verification system known as the Land surface Verification Toolkit (LVT; Kumar et al. (2012)), en-
abling the systematic verification and evaluation of modeled land surface states against independent
measurements and datasets. LVT-based evaluations are used in this study to evaluate the utility of
VOD assimilation approaches.

### 2.3   Data assimilation configuration

Similar to the assimilation strategy employed in Kumar et al. (2019b), a 1-d Ensemble Kaman Filter
(EnKF; Reichle et al. (2002)) method is used for the assimilation of VOD retrievals. The EnKF
algorithm works with an ensemble of model states, which is propagated forward in time using the
LSM and updated toward the observation based on the relative uncertainty of the model states and
the observation. The model state update at time $k$ is represented by the following equation:

$$x_k^{i+} = x_k^{i-} + \mathbf{K}_k \left[ y_k^i - \mathbf{H}_k x_k^{i-} \right], \tag{3}$$

where $x_k^{i-}$ and $x_k^{i+}$ represents the model state for the $i^{\text{th}}$ ensemble member before and after the up-
date, respectively. The observation vector is represented by $y_k$ which is connected to the model states
through the observation operator $\mathbf{H}_k$. The relative weight given to the innovations ($\left[ y_k^i - \mathbf{H}_k x_k^{i-} \right]$)
in the analysis update is determined by the Kalman gain term ($\mathbf{K}_k$).

As described in Kumar et al. (2019b), the innovations in the LAI DA configuration are specified
by comparing the model prognostic LAI variable with the observations. The $y_k$ in this case is the
remotely sensed LAI and $\mathbf{H}_k x_k^{i-}$ is the model's LAI estimate. In case of VOD assimilation, the
computation of the innovations is tricky as Noah-MP does not directly estimate VOD within the
model. To overcome this limitation, the VOD observations are rescaled into the LAI space in the data
assimilation configuration. The rescaling is performed using a seasonally varying CDF-matching
(Kumar et al. (2015)) and by using the MODIS-based LAI observations from the Global Land Cover


Facility (GLCF) Global LAnd Surface Satellites (GLASS; Xiao et al. (2016)) project at University of
Maryland as the LAI reference. The MODIS-based LAI retrievals from the GLASS LAI product are
generated using a general regression neural network approach, enabling a spatially and temporally
continuous record of LAI available at 8-day intervals on a $0.05°$ regular latitude-longitude global
grid. We use GLASS data as the LAI reference, due to the improved spatio-temporal coverage as
well as the high quality of the product established in intercomparison studies (Liao et al. (2012); Fang
et al. (2013); Xiao et al. (2016)). Monthly CDFs are computed for both the VOD and LAI datasets
using all available data, at every model grid point. For example, the LPRM X-band and C-band
CDFs are computed using datasets from 2002-2018 whereas SMAP CDFs are computed using the
available data from 2015 to 2019. To increase the sampling density in the SMAP CDF calculations,
a spatial sampling window of 2 pixels is employed. The GLASS LAI CDFs are computed using a
time period of 2000 to 2018.

The rescaling is performed with the assumption that there is a strong correlation between VOD
and LAI. The use of VOD as an analog to existing vegetation measurements such as optical-infrared
indices and fluorescence has been suggested in prior studies (Konings et al. (2017)). For exam-
ple, Albergel et al. (2018) demonstrated that the modeled LAI and VOD derived from the C-band
backscatter measurements from the ASCAT sensor had high correlations over most of the CONUS.
Figure 2 presents a similar comparison, where maps of the correlation of X-, C-, and L-band re-
trievals of VOD against the MODIS-based LAI retrievals from the GLASS project are shown. Based
on the mutual availability of datasets, the correlation maps are generated using a time period of 2002-
2018 and 2015-2018 for the LPRM and SMAP comparisons, respectively. Strong correlations are
215    observed in the LPRM X-band VOD vs LAI comparisons in most parts of the domain except over
the arid, southwest region of the U.S. The agreements between the LPRM X-band VOD and LAI are
particularly strong over the eastern U.S., agricultural areas of the Midwest, central California val-
ley, which are regions of high vegetation density. The level of agreement between VOD and LAI is
weaker in the C-band and L-band comparisons compared to the X-band. This is consistent with the
220    fact that the attenuation of the lower frequency measurements from vegetation is less compared to
that for X-band. The documented influence of RFI contamination over CONUS (Njoku et al. (2005))
is also evident in the C-band comparisons. Interestingly, the SMAP-based L-band retrievals of VOD
show stronger correlations with LAI than those from C-band, particularly in the eastern U.S. This is
a function of data from different sensing platforms, the use of different retrieval algorithms, and dif-
225    ferent data record lengths. As documented in prior studies, the high frequency VOD measurements
are more sensitive to the top of the vegetation (Konings et al. (2017)). The L-band measurements, on
the other hand, are more representative of the vegetation changes in the deeper layers of the canopy.
The strong relationship between VOD and LAI observed in Figure 2 confirms that the rescaling
procedure used in the DA configuration is reasonable.




This article also compares and contrasts the impact of assimilating VOD with that from incorporating soil moisture retrievals from the L-band microwave instruments. Soil moisture in the LSMs is a model-specific quantify, an index of the moisture state (Koster et al. (2009)). As a result, there are significant differences in soil moisture estimates from different LSMs, even when forced with the same meteorology and land surface parameters (Dirmeyer et al. (2006)). Similarly, remote sensing based estimates of soil moisture are also indirect measurements generated through a retrieval model from direct measurements of the microwave emission of the land surface. Here we apply the commonly used strategy of CDF-matching (Reichle and Koster (2004) to address the relative differences between the remote sensing and LSM-based soil moisture by rescaling the soil moisture retrievals into the LSM climatology before assimilation. The CDFs are computed separately at each grid point on a monthly basis. Note that such a configuration only incorporates the anomaly information in the soil moisture retrievals and ignores the information inherent in the mean soil moisture signals (Kumar et al. (2015)). Similar to the strategy used in prior studies, soil moisture retrievals are excluded near water bodies, for being at the edge of the swath, when soil is frozen/covered by snow, and when the vegetation cover is thick (Kumar et al. (2019a)), to account for the known limitations of passive microwave-based soil moisture retrievals.

An ensemble size of 20 is used in the data assimilation integrations, with perturbations applied to a number of meteorological fields and the model state vector to develop representations of model uncertainty. Based on the settings used in recent DA studies in the NLDAS-2 configuration (Kumar et al. (2019a, b)), the precipitation ($P$) and downward shortwave radiation ($SW$) fields are perturbed with multiplicative perturbations with mean of 1 and standard deviations of 0.3 and 0.5, respectively. Further, additive perturbations with mean zero and standard deviation of $50\ W/m^2$ are applied to the downward longwave radiation ($LW$) fields. The hourly forcing perturbations also include cross correlations ($\rho$) between the forcing variables, with values of $\rho(SW, P) = -0.8$, $\rho(SW, LW) = -0.5$, and $\rho(LW, P) = 0.5$. For VOD DA, additive perturbations with a standard deviation of 0.01 are applied to the model LAI fields (Kumar et al. (2019b)), every 3 hours. The updated LAI from DA is divided by the specific leaf area to revise the leaf biomass variable within Noah-MP. The state vector used in the soil moisture DA consists of the top soil moisture layer of Noah-MP, which is perturbed with an additive noise of $0.02\ m^3/m^3$, applied every 3 hours. The perturbations also include time series correlations employed through a first order autoregressive (AR(1)) model with timescales of 24 and 3 hours, for the forcing and model state variables, respectively. The input observation error standard deviation is set to $0.04\ m^3/m^3$ for assimilating SMAP soil moisture retrievals, whereas the observation error standard deviation is set to 0.05 for the scaled VOD retrievals, based on settings from recent studies employing soil moisture (Kumar et al. (2019a)) and LAI (Kumar et al. (2019b)) retrievals. The assimilation of each dataset is performed in a sequential manner, based on their respective measurement or overpass times.



## 3 Results

This section presents an evaluation of the impact of assimilating VOD retrievals on key terrestrial water and carbon states and fluxes. The impact of assimilating the X-band and C-band VOD retrievals is presented first, followed by the evaluation of assimilating L-band VOD retrievals from SMAP. Since

soil moisture is typically considered the primary retrieval from microwave remote sensing, we also evaluate the relative benefits of assimilating both SMAP surface soil moisture and VOD retrievals. The impact of DA is quantified by comparing to a large suite of reference measurements of soil moisture, evapotranspiration, gross primary productivity (GPP), and streamflow. The evaluations are conducted using the NASA Land surface Verification Toolkit (LVT; Kumar et al. (2012)).

### 275  3.1  Impact of assimilating X-band and C-band VOD retrievals

The impact of assimilating VOD retrievals on the simulated ET estimates is shown in Figure 3, which shows the change in RMSE and correlation (R) of ET in the DA simulation relative to the OL. These evaluation metrics are computed using two reference data products: (1) the gridded 0.5 deg, monthly FLUXNET multi-tree-ensemble (MTE) product based on tower ET measurements (Jung

et al. (2009); available from 1982-2008) and (2), the 4 km, daily Atmosphere-Land Exchange Inverse (ALEXI; Anderson et al. (2007)) model product, developed using TIR measurements, available from 2001 onwards. Strictly speaking, ALEXI is a model product with associated biases and errors of its own. Comparatively, FLUXNET MTE can be considered as a close analog to a true ground-reference product, since it is derived by empirically upscaling eddy covariance measurements, though it is also

affected by the sampling density and consistency of site measurements. Therefore, RMSE is used as the metric of evaluation in the FLUXNET MTE comparison, whereas R is used to assess the improvements in ET from DA relative to ALEXI. Figure 3 indicates that the assimilation of VOD generally provides beneficial impacts on ET, consistently in the comparisons against both reference datasets. In addition, most prominent improvements are obtained over the agricultural areas over

the Midwest U.S., lower Mississippi basin, the central California valley, and parts of Mexico. Prior studies have documented that ALEXI is particularly skillful in detecting spatial features from agricultural management impacts (Hain et al. (2015)). The fact that the spatial patterns of improvements in ET in Figure 3 is well correlated with the crop areas provides added confirmation that VOD assimilation is helpful in improving the representation of managed vegetation (as noted in Kumar et al.

(2019b)).

The impact of VOD assimilation on the carbon fluxes is assessed by focusing on GPP, which represents the total carbon fixation through photosynthesis. The model simulated GPP is compared against two datasets: (1) gridded 0.5 deg estimates of GPP from the FLUXCOM project (Tramontana et al. (2016); Jung et al. (2017)) and (2) remote sensing retrievals of solar induced fluorescence

(SIF) from the Global Ozone Monitoring Experiment-2 (GOME-2) aboard the MetOp-A satellite



(Joiner et al. (2014); Guanter et al. (2014)). Similar to FLUXNET MTE, the FLUXCOM estimates are produced by upscaling point measurements using machine learning approaches. SIF, which is a measure of the re-emission of light during photosynthesis, is considered an observational analog of GPP. Figure 4 provides an evaluation of GPP against these two reference datasets using two different

metrics. Compared to FLUXCOM, the improvements in RMSE from the X-band and C-band VOD are shown in the top panel of Figure 4. The bottom panel of Figure 4 show the improvements in correlation (R) of GPP against the GOME-2 SIF measurements from VOD DA. These independent comparisons against two different products further confirm the beneficial role of VOD DA over the agricultural regions, similar to the patterns in the ET comparisons of Figure 3. The RMSE of simu-

lated GPP is reduced and the correlation with the SIF retrievals is improved through the assimilation of VOD.

Figures 3 and 4 also offer assessments of the relative utility of X-band and C-band VOD retrievals. While the spatial patterns of improvements are generally similar in both X- and C-band assimilation configurations, the assimilation of X-band VOD provides stronger improvements. This

is consistent with the fact that the attenuation of the microwave signal reduces for lower frequency measurements. As both Figures 3 and 4 indicate the strong influence of vegetation type in the improvement maps of ET and GPP, we quantify the domain-averaged percentage improvements by vegetation type, shown in Figure 5. For simplicity, we use a simpler vegetation classification scheme by grouping the Evergreen Needleleaf, Broadleaf Needleleaf, Deciduous Needleleaf, and Deciduous

Broadleaf forests into a "Forest" category, the Mixed Forests, Woodlands, and Wooded Grasslands into a "Mixed Forests" category, and Closed Shrublands and Open Shrublands into a "Shrublands" category. Note also that the percentage improvements shown in Figure 5 are for different metrics. For FLUXNET and FLUXCOM comparisons, the percentage improvements are shown for RMSE, whereas for ALEXI and GOME-2, the percentage improvements in R are shown. Figure 5 confirms

that the largest impact of VOD assimilation is over Croplands, providing up to a domain-averaged improvement of 10% and 38% in ET and GPP, respectively. Significant improvements are also observed over areas with moderate vegetation such as Grasslands and Shrublands, while over Forests and Mixed Forests, the level of improvements reduces. Over Bare soil areas, the impact of VOD assimilation is very small, due to the lack of vegetation influence on ET and GPP. As seen in Fig-

ure 5, compared to X-band VOD-DA, the level of improvements with C-band VOD-DA reduces. For ET, at a domain averaged scale, the assimilation of C-band and X-band VOD retrievals provide 4.6 and 6.8% improvements in RMSE, respectively, when compared to FLUXNET MTE. Compared to ALEXI, C-band VOD DA provides 3.1 (2.0)% domain-wide improvements in RMSE (R) of ET, respectively. These percentage improvements in RMSE(R) increase to 4.0 (2.7)% for X-band VOD

assimilation. Similarly, the domain averaged percentage improvement in RMSE of GPP with C-band VOD assimilation is 17.3 and it improves to 22.3 with X-band VOD assimilation. The domain aver-





aged correlation of the OL-based GPP with GOME-2 SIF is 0.53 and it improves to 0.62 and 0.66 with C-band and X-band assimilation, respectively.

The impact of VOD assimilation on other land surface states such as soil moisture, terrestrial water storage, and streamflow is also evaluated using a number of reference products. The in-situ measurements from the International Soil Moisture Network (ISMN; Dorigo et al. (2011)) are used for evaluating soil moisture fields. Similar to the Kumar et al. (2019b) study, hourly data from 934 stations from 9 different networks within ISMN is used for evaluating the soil moisture estimates. As it is well known that model simulated soil moisture and in-situ measurements are significantly biased relative to each other, the soil moisture evaluations are performed using the anomaly correlation metric. The surface and root zone soil moisture anomalies are computed as the differences between the daily soil moisture and the respective monthly mean values.

Overall, VOD assimilation has marginal impacts on the simulated soil moisture estimates. The domain averaged anomaly R values for the OL surface and root zone soil moisture are 0.54 and 0.47, respectively. With the C-band assimilation, these values marginally improve to 0.55 and 0.48, respectively. Similarly, the X-band assimilation also lead to domain averaged anomaly R values of 0.55 for surface soil moisture and 0.49 for root zone soil moisture. Though these domain averaged changes from assimilation are not statistically significant, there are larger regional improvements, particularly for the root zone estimates. Notably, regional improvements are observed over the central plains and the lower Mississippi regions (not shown), consistent with the spatial patterns seen in the ET and GPP evaluations.

The simulated TWS anomalies are also evaluated against the Gravity Recovery and Climate Experiment (GRACE) satellite-based Tellus product ((http://grace.jpl.nasa.gov/data/gracemonthlymassgridsland/), available on 1° horizontal resolution grids (Landerer and Swenson (2012)), during the lifespan of the mission (2003-2017). The domain averaged anomaly R for the OL-based TWS is 0.45, and it improves to 0.48 with C-band and X-band VOD assimilation. These improvements are statistically significant. In addition, larger improvements in anomaly R (as high as ∼0.28) are observed over the agricultural areas of Central Plains and central California (not shown).

### 3.2 Comparing assimilation of optical sensor-based LAI and passive microwave-based VOD

The impact of passive microwave based VOD assimilation relative to assimilating LAI retrievals from optical instruments is presented in Table 1. The percentage improvements in various terrestrial water and carbon components against reference datasets, from the assimilation of MODIS LAI (from the Kumar et al. (2019b) study) and the X- and C-band VOD retrievals are presented in this table. Note that both Kumar et al. (2019b) and the current study use the exact same model configuration, land surface parameters and boundary conditions. Overall, the magnitude of improvements from VOD assimilation is comparable to that of assimilating LAI. The LAI assimilation has no impact on the aggregate soil moisture skill, whereas the VOD assimilation marginally improves the





domain averaged anomaly R values for both surface and root zone soil moisture. Both VOD and LAI assimilation improves ET estimates, with the percentage improvements in RMSE ranging from 3-

7%, depending on the reference dataset used. Overall, comparable improvements in ET are obtained with X-band VOD DA and LAI DA, with ET estimates from C-band VOD DA being marginally less skillful than those from LAI DA. Both LAI and VOD assimilation provides significant improvements (with approximately 17-24 % domain averaged improvements) in GPP. Similar to the changes in soil moisture, marginal improvements in TWS and streamflow are obtained from both VOD and LAI as-

similation. Note that though the magnitude of added improvements is small for certain variables, larger regional improvements are observed in these comparisons.

Overall, the comparison in Table 1 confirms that VOD DA is an effective option for incorporating remote sensing-based inputs of vegetation conditions. Note that the spatial resolution of passive microwave retrievals is typically coarser than those from the optical/IR sensors. In addition, passive

microwave measurements are only available from low earth orbits (LEO) due to the antenna size requirements, whereas instruments on geostationary orbits provide increased sampling frequency from optical/IR instruments. Since the results suggest that assimilation of passive microwave-based VOD retrievals provide comparable skill to that from optical sensor-based LAI, assimilation of both types of datasets will allow minimizing the sensing, coverage, and spatial resolution-based limitations of

each sensor.

### 3.3 Impact of assimilating L-band VOD retrievals from SMAP

In this section, the impact of assimilating L-band VOD retrievals from SMAP is evaluated and is contrasted with corresponding improvements obtained with higher frequency VOD assimilation. As SMAP data availability is limited to 2015 April - present, all evaluations in this section are limited

to 2015 April - 2018 December. Note that not all reference datasets used in Section 3.1 are available during this limited time period.

Figure 6 quantifies the impact of assimilating L-band VOD retrievals from SMAP on ET and GPP. Similar to the results seen with the X- and C-band VOD assimilation, SMAP VOD DA also provides systematic improvements in the simulated ET and GPP, comparable to those from X-band VOD as-

similation. The patterns of improvements in ET in the ALEXI comparison are similar to those in Figure 3. Strong improvements in ET and GPP over the corn and soybean areas of the Midwest and lower Mississippi are observed in the SMAP VOD DA evaluations. The ALEXI comparison indicates that the assimilation of VOD retrievals also improves the simulation of ET over the Southeast U.S., an area with thick vegetation density. Similar patterns are seen in the comparisons to GOME-2

SIF, where significant improvements in the correlation of simulated GPP with SIF observations are obtained over the Southeast U.S. and agricultural areas of the Midwest. These results suggest that the significant utility of the VOD retrievals are over the agricultural areas and locations with strong

vegetation seasonality. Note that the patterns in Figures 3 and 6 are not exactly equivalent due to the different time periods used in the evaluations.

To further examine the impact of VOD DA, Figure 7 shows the time series of VOD, rescaled VOD (using CDF-matching) as LAI, and the corresponding change in ET in DA simulations (relative to OL) at two locations. Location A is in Iowa with cropland as the dominant landcover and location B is in Montana with grassland as the dominant landcover (Figure 1). The cropland location is used as an analog of an area where agricultural activity is likely present, whereas the grassland location

is representative of a region where the natural variability is the dominant factor in the vegetation and ET seasonality. Note also that at location A, large improvements in ET and GPP are observed, whereas at location B, only marginal improvements are noticed in ET and GPP.

Over the cropland location A, both the L-band and X-band VOD estimates are consistent with each other, in terms of the amplitude and seasonality. The peak VOD seasonality is in the late summer and

early fall, which is reflected in the rescaled LAI estimates. The model OL based LAI, on the other hand, has an earlier peak, in the summer months. The assimilation of the rescaled VOD estimates leads to corrections in both the magnitude and phase of the LAI relative to the OL estimates. This also leads to a corresponding phase shift and increase in the peak ET estimates from DA. The changes in the ET in the DA simulations over location A ranges from approximately -30 to 40 W/m$^2$ during the

summer and fall months.

Compared to location A, over the grassland location B, there are small climatological differences in the VOD retrievals from X- and L-band. Since seasonality in the anomalies and not the mean signal is the key factor in CDF matching, the rescaled X- and L-band VOD estimates are similar to each other. Overall, the changes in LAI in the assimilation runs relative to the OL are small, likely

because this is an area with sparse vegetation. In year 2017, the main impact of DA is to increase the amplitude of LAI, whereas in 2018, the LAI estimates in the DA and OL are fairly consistent, except for a small phase shift. In the summer and fall months, the assimilation leads to approximately $\pm$ 10 W/m$^2$ changes in ET. The independent evaluations of ET in Figure 3 confirm that these phase and magnitude corrections in LAI through the VOD DA (particularly at location A) are accurate. Similar,

but more muted impacts relative to the X-band DA are seen from the C-band DA (not shown).

### 3.4 Comparison of soil moisture and VOD DA

As there is a long legacy of retrieving soil moisture from microwave radiometry, the key focus of the associated missions and data assimilation studies has been on evaluating and demonstrating the utility of retrieved soil moisture measurements (Reichle et al. (2007); Liu et al. (2011a); Draper et al.

(2012); Hain et al. (2012); Kumar et al. (2014); Lievens et al. (2017)). These studies demonstrate the potential of remote sensing soil moisture retrievals to improve the simulation of moisture states. Efforts to translate the improvements in the soil moisture states to other water and energy stores, on the other hand, have only reported marginal success. Though changes in soil moisture states from


DA impacts the land-atmosphere fluxes at diurnal temporal scales (Santanello et al. (2016)), their

impacts at broader spatial and temporal scales are small. For example, studies at continental scales such as Peters-Lidard et al. (2011) and Martens et al. (2016) reported minor impacts in the simulated ET estimates from the assimilation of LPRM soil moisture retrievals. Here we compare and contrast the relative utility of assimilating the soil moisture and VOD retrievals from SMAP on various water and carbon states.

Figures 8 to 10 show the impacts of assimilating SMAP soil moisture and VOD retrievals on various land surface water and carbon states. Using the in-situ soil moisture measurements from ISMN as the reference, Figure 8 shows the changes in anomaly R of surface and root zone soil moisture from soil moisture and VOD assimilation. Overall, soil moisture DA has a significant and positive impact on the simulation of surface soil moisture, particularly in the Western U.S. and

Highplains. The impact of soil moisture DA over the Eastern U.S. is small, as these regions of high vegetation density are generally excluded from soil moisture DA. Comparatively, VOD assimilation has little impact on surface soil moisture, as the changes in anomaly R are not statistically significant in most locations. Both soil moisture and VOD assimilation have stronger impacts on root zone soil moisture estimates. The assimilation of SMAP soil moisture improves the root zone estimates over

the lower Mississippi and parts of the Western U.S. including California, Nevada, and Colorado. The patterns of improvements and degradations in root zone soil moisture are more mixed in the VOD assimilation results, over these same areas.

Figure 9 shows the impact of soil moisture assimilation on ET and GPP. Consistent with prior studies, the impact of soil moisture assimilation on ET and GPP is small over most of the domain.

Compared to ALEXI, SMAP soil moisture assimilation marginally improves the correlation of simulated ET over parts of central California, Washington, Montana, Texas and lower Mississippi, with small degradations over several Western States. The SMAP soil moisture assimilation has little impact on the simulation of GPP, as the change map of R against the GOME-2 SIF measurements shows no distinct spatial patterns of improvements or degradations. Comparatively, VOD assimilation has

a strong and mostly beneficial impact on the simulation of ET and GPP, as shown in Figure 6. In the comparisons against ALEXI and GOME-2, strong patterns of improvements are observed over the agricultural areas of the U.S. such as the central Plains, lower Mississippi basin and central California, from VOD DA.

Finally, the impact of both soil moisture and VOD assimilation on streamflow is captured in Fig-

ure 10 which shows the normalized improvements in Nash Sutcliffe Efficiency (NSE) of streamflow using the daily USGS measurements as the reference. These locations represent areas where the streamflow measurements are minimally impacted by reservoir operations (Kumar et al. (2014, 2019b)). The normalized NSE improvements are represented using the Normalized Information Contribution (NIC) metric (Kumar et al. (2014)), with positive and negative NIC values indicating

benefit and degradation from assimilation, respectively. Soil moisture assimilation has a beneficial



impact on the streamflow simulation with improvements over the Midwest and Eastern U.S. and degradations over the Southeast and parts of the Missouri basin and Western locations. The impact of VOD assimilation on streamflow is marginal and is mostly restricted to the Midwest areas, which also are correlated with the corn growing areas. Note that though there are some regional patterns
of improvements are degradations in streamflow from soil moisture or VOD DA, these changes are small (most of the NIC changes are in the range of +/- 0.05 to 0.02).

To further investigate the relative utility of VOD and soil moisture DA, we compare the time series of changes in surface soil moisture, ET, transpiration and bare soil evaporation at two locations (C and D) in the domain, in Figure 11. Location D is in the eastern U.S., representing a wet region
with thick vegetation whereas location C is in the arid western U.S. with moderate vegetation. In the arid location C, soil moisture DA leads to changes in surface soil moisture primarily in the summer months, with differences as large as 0.05 m3/m3 relative to the OL. The changes in soil moisture subsequently drives the changes in ET estimates. The comparison of the time series of transpiration and bare soil evaporation indicates that the changes in ET at location C are more directly connected
to the changes in bare soil evaporation. There is essentially no change in transpiration from soil moisture DA at this location, but larger changes in bare soil evaporation occur as a result of changing soil moisture. Comparatively, at location C, VOD DA has little impact on soil moisture and ET. The changes in LAI introduced by VOD DA lead to a small increase in transpiration and a minor reduction in bare soil evaporation. These changes in the evaporative fluxes are not driven by the soil
moisture changes, rather by the small change to the vegetation coverage.

In contrast, over location D, there are little changes in soil moisture and ET from soil moisture DA, because not many observations are assimilated over this area with thick vegetation. The time series of transpiration and bare soil evaporation confirms that soil moisture DA has little impact on the evaporation regime. VOD DA, on the other hand, leads to large changes in ET as a result of the
changes in LAI. The increased LAI leads to increased transpiration and root uptake of soil moisture. The reduction in root zone soil moisture also leads to reduced bare soil evaporation. Overall, VOD DA leads to increased ET in the summer months at this location because of these changes. These comparisons indicate that there is independent information in both soil moisture and VOD retrievals of SMAP that is useful in improving estimates of ET. Soil moisture information is more impact-
ful over water limited regions, where moisture conditions on the land are the primary controls on the evaporative fluxes. Over energy-limited domains with thick vegetation, vegetation growth and stomatal control, more than surface moisture conditions, influence the ET evolution. Since passive microwave retrievals of soil moisture are unreliable over such areas, the use of VOD provides an effective alternative. The above cases show a direct impact on the relative importance of transpi-
ration vs bare soil evaporation in the ET generation. Accurate estimation of this ET partitioning is important for a proper connection to the carbon cycle (Kumar et al. (2018)).



The small improvements in hydrological budget terms such as ET and streamflow from soil moisture DA are also partly due to the mechanisms used in soil moisture DA configurations. As noted earlier, because of the use of rescaled retrievals (using CDF matching) in soil moisture DA, the analysis updates only reflect the corrections in the anomalies of soil moisture, rather than large changes in mean soil moisture estimates. The transformed VOD retrievals, on the other hand, are ingested directly as LAI within the LSM, essentially allowing the incorporation of the information inherent in the mean VOD/LAI signals. The limited use of the information in the soil moisture DA configuration is partly the reason for the limited impact on water budget states such as ET.

## 3.5 Joint assimilation of soil moisture and VOD retrievals

As the results in the previous section indicate that assimilation of soil moisture and VOD can provide mutually exclusive information, an assimilation configuration that employs these retrievals simultaneously is developed. Similar to the univariate configurations, in this multivariate configuration, soil moisture retrievals are used to update the surface soil moisture state, whereas VOD retrievals are used to update the prognostic LAI variable within the LSM. Figure 12 summarizes the impact on key water budget terms as a result of the joint assimilation of soil moisture and VOD. Overall, the joint assimilation consolidates the beneficial impact from the univariate assimilation configurations. For example, the multivariate DA configuration provides improved skills in both surface and root zone soil moisture, whereas the univariate VOD DA has little impact on surface soil moisture. Similarly, the univariate soil moisture DA configuration has little influence on the ET skill, whereas the ET improvement maps from the joint assimilation mirrors the patterns of changes obtained with univariate VOD DA. The spatial influence of the individual assimilation configurations is also evident in these comparisons. For example, the ET improvement map (with ALEXI as the reference) from the joint DA shows strong patterns of improvements in the eastern U.S. similar to the result from the VOD DA configuration. The improvement in ET is accompanied by even higher percentage improvements in GPP. It is interesting to note the strong improvements centered on the Mississippi, as in where partitioning contributes to ET uncertainty (Kumar et al. (2018)). In the western U.S., there are some patterns of degradation in ET, similar to what is observed when assimilating soil moisture alone. Similarly, in the streamflow comparisons, the joint assimilation shows strong patterns of improvements in areas east of Mississippi, whereas the impact of assimilation is mostly disadvantageous in the western parts of the domain. As noted earlier, these patterns reflect the larger impact of VOD and soil moisture in the energy (eastern U.S.) and water limited (western U.S) domains.

## 4 Summary

Vegetation conditions have a significant influence on the terrestrial water, energy, and carbon exchanges and feedbacks. Through stomatal control, plants influence transpiration, root uptake of soil


moisture, and evaporative fluxes. The presence of vegetation also impacts the evolution of snow by influencing surface albedo and the amount of net radiation on the land surface. In addition to the changes in vegetation phenology driven by natural variability, anthropogenic activities such as agriculture and vegetation disturbances also significantly alter the vegetation characteristics on the land

surface. Data assimilation of remotely sensed estimates of vegetation conditions within land surface models enable the refinement of modeled estimates, enhancement of the spatio-temporal coverage of remote sensing measurements, and the extension of the remote sensing vegetation information to water, energy, and carbon states and fluxes.

Remote sensing based estimates of vegetation conditions are typically developed from multi- and

hyperspectral optical and thermal satellite sensors. Though passive microwave sensors are often used for retrieving soil moisture estimates, they also enable the estimation of vegetation optical depth, an index of above-ground canopy biomass. As microwave measurements are not influenced by clouds, they can be made in virtually all weather conditions. This article examines the utility of VOD retrievals from passive microwave sensors by assimilating them within the dynamic phenology

model of Noah-MP LSM.

The study is conducted in the NLDAS-2 configuration over the Continental U.S. A suite of VOD retrievals from X-, C- and L-band instruments is assimilated in Noah-MP using a 1-d ensemble kalman filter algorithm. The X-and C-and retrievals from the Land Parameter Retrieval Model, whereas the L-band retrievals of VOD are from SMAP. Since Noah-MP does not include a prog-

nostic representation of VOD, the assimilation is conducted by transforming the VOD retrievals into LAI estimates, using the MODIS-based GLASS LAI product. The impact of assimilating VOD on key water and carbon budget terms is evaluated by comparing against a large suite of reference datasets.

The assimilation of VOD from the passive microwave sensors is found to have a significant ben-

eficial impact on improving the simulation of ET and GPP, particularly over the agricultural areas of the U.S. The assimilation of X-band based VOD retrievals is found to provide larger improvements in ET, relative to the assimilation of C-band VOD retrievals. Though beneficial, the impacts on soil moisture, terrestrial water storage, and streamflow from VOD DA are marginal. Regionally, the largest impacts on these variables are also observed over the agricultural areas. Though the time

period of available data is limited, the assimilation of L-band VOD retrievals from SMAP is also found to have significant beneficial impacts on the simulation of ET and GPP, similar to that from the X-band VOD DA.

Though passive microwave-based measurements are available in nearly all-weather conditions, their spatial resolution and temporal frequency is coarser than the optical/IR based vegetation esti-

mates. This study compared the impact of VOD assimilation to that of assimilating optical sensor-based LAI from a prior study. Overall, the magnitude of improvements from VOD DA is comparable to that from assimilating MODIS LAI. These findings confirm that assimilation of VOD retrievals



can provide an effective augmentation or alternative to assimilating data from optical sensors, enabling the mitigation of sensing, coverage, and spatial resolution-based limitations of each type of

sensor.

The relative and joint utility of assimilating soil moisture and VOD retrievals from SMAP are also examined in this study. Overall, the assimilation of soil moisture retrievals has a positive impact on the simulation of surface soil moisture and little impact on evaporative fluxes. In contrast, VOD DA has significant impacts on the simulation of vegetation conditions, root zone soil moisture, and

evapotranspiration. Over water limited domains with sparse vegetation where soil moisture is the primary control on ET, the assimilation of surface soil moisture is more beneficial than VOD DA. Over regions with dense vegetation and where water availability is not limiting, transpiration has a significant influence on evapotranspiration. The assimilation of VOD is more beneficial in developing improvements in ET over such locations. In addition, when vegetation coverage is dense, the soil

moisture retrievals have large uncertainty and are unreliable. In those areas, the use of VOD provides an alternate way to develop improved estimates of terrestrial hydrologic responses informed by remote sensing. The results in the paper also confirm that the soil moisture and VOD retrievals provide independent information that can be jointly exploited through their simultaneous assimilation.

As noted in the description of the data assimilation methodology, the VOD retrievals are assimi-

lated by rescaling them to the GLASS MODIS LAI climatology. This approach was employed as the prior study Kumar et al. (2019b) demonstrated significant positive impacts from the assimilation of the GLASS LAI data. Such an approach is needed also because the LSM does not have a prognostic representation of VOD. Though the beneficial impacts observed in the results indicate that this is a reasonable strategy, the rescaling essentially ignores the information on vertical heterogeneity in

the canopy from these sensors. For example, the X-band data is documented to be more sensitive to the vegetation, whereas the L-band data is more representative of the lower canopy. A more direct use of the VOD data is likely to help in resolving these sensitivities within modeling. Extensions to this study that either uses a prognostic representation of VOD or a forward model that simulates VOD will enable such approaches. The current study serves as a useful benchmark for such future

extensions.

Finally, as noted earlier, NLDAS-2 configuration is a conservative environment to evaluate the utility of data assimilation configurations due to availability of high quality boundary condition data. The significant utility of VOD DA demonstrated in this paper suggests that larger benefits from VOD DA are likely over areas with lower quality meteorological boundary conditions.

*Acknowledgements.* Funding for this work was provided by the NOAA's climate program office (MAPP program). Computing was supported by the resources at the NASA Center for Climate Simulation. The NLDAS-2 forcing data used in this effort were acquired as part of the activities of NASA's Science Mission Directorate,



and are archived and distributed by the Goddard Earth Sciences (GES) Data and Information Services Center (DISC).



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



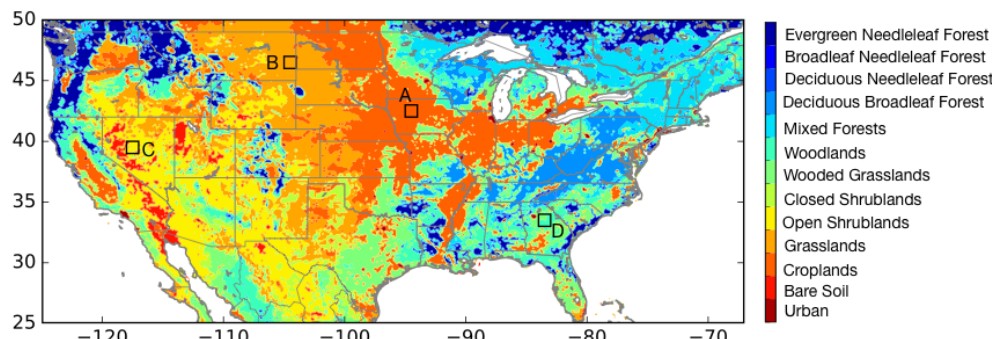

**Figure 1.** Map of the modeling domain with the UMD landcover classification as the background. The locations A, B, C, and D denote the areas used for time series comparisons to examine the impact from VOD DA.

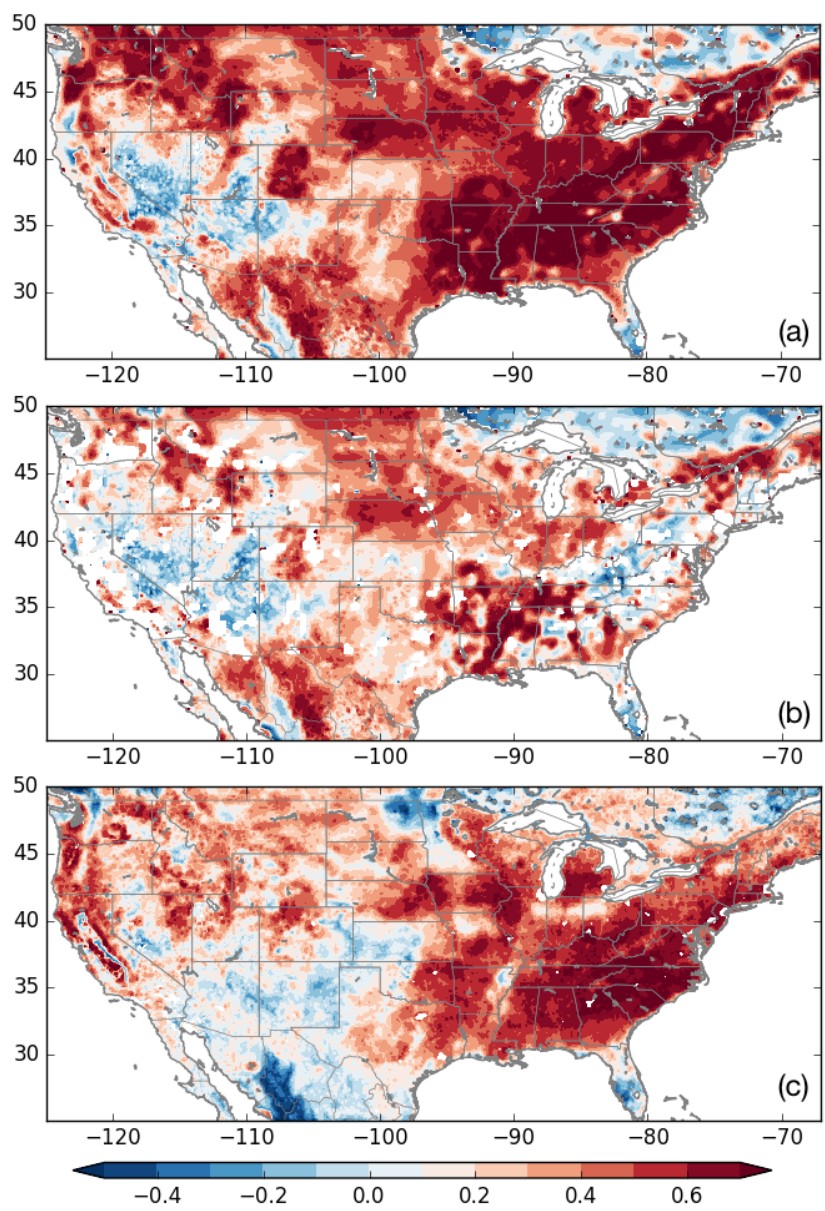

**Figure 2.** Correlation of VOD retrievals from LPRM X-band (a), LPRM C-band (b), and SMAP L-band (c) against the MODIS-based LAI retrievals. The LPRM and SMAP comparisons employ data during 2002-2018 and 2015-2018 time periods, respectively.



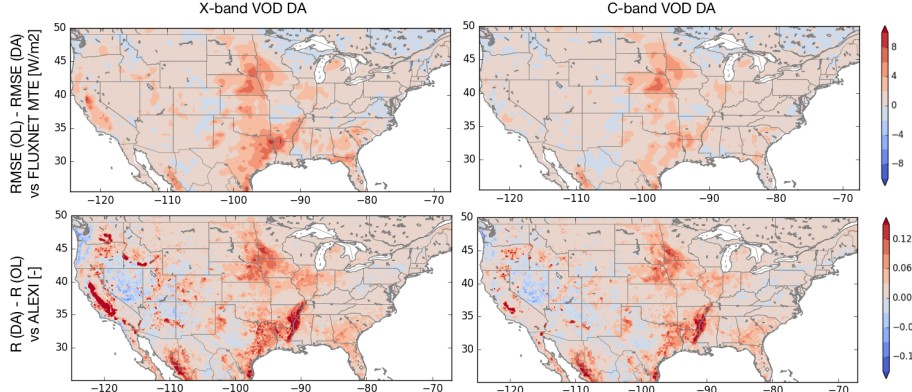

**Figure 3.** RMSE differences (W/m2) of evapotranspiration from X-band VOD (left column) and C-band VOD (right column) assimilation relative to the OL integration, using two reference dataset. The time periods in the RMSE comparisons are 2000-2008 and 2000-2018, for FLUXNET MTE and ALEXI, respectively. The warm and cool colors represent the increase and decrease in RMSE due to VOD DA, respectively.

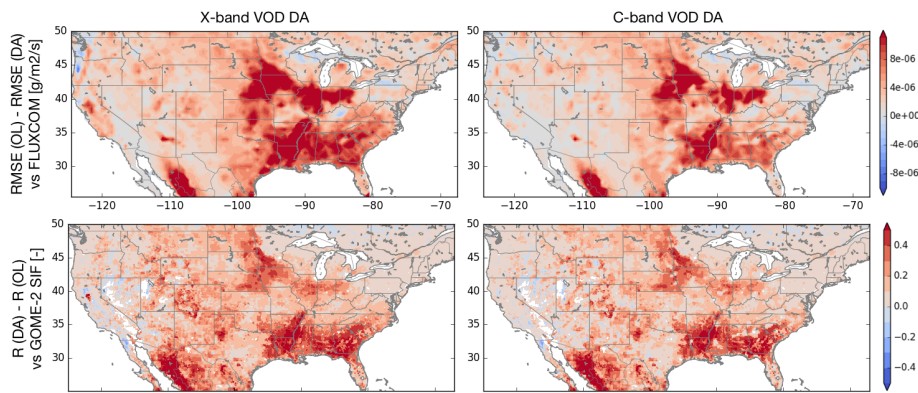

**Figure 4.** Changes in RMSE of GPP (expressed as RMSE(OL) - RMSE (DA)) in units of $gm^{-2}s^{-1}$ using the FLUXCOM data as the reference (top row) and R of modeled GPP with solar induced fluorescence data from GOME-2 (bottom row), expressed as R (DA) - R (OL). The warm colors represent improvements from DA and cool colors represent degradations resulting from DA.



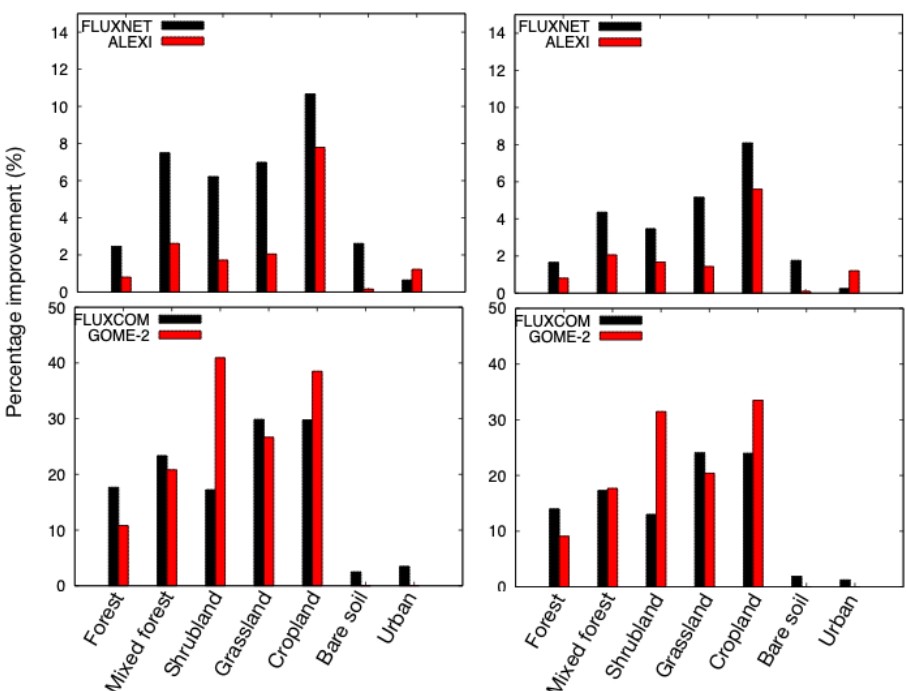

**Figure 5.** Domain-averaged percentage improvements in ET (top row) and GPP (bottom row) stratified by vegetation type. The left column represents the impact of DA from X-band VOD whereas the right column represents the impact of C-band VOD DA. The percentage improvements in ET using FLUXNET and ALEXI reference datasets are expressed for RMSE and R metrics, respectively. Similarly, the percentage improvements in GPP using FLUXCOM and GOME-2 SIF are for RMSE and R metrics, respectively.

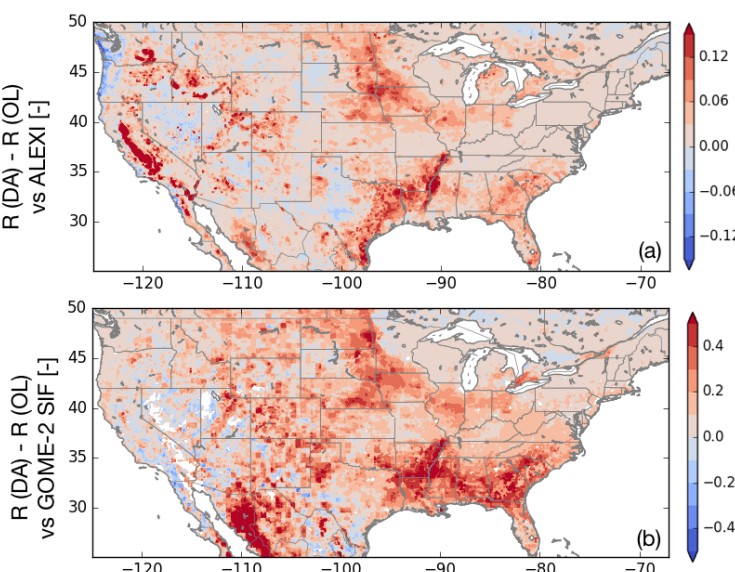

**Figure 6.** Changes in the skill of simulated evapotranspiration and GPP as a result of assimilating L-band VOD estimates from SMAP. The top panel represents the changes in RMSE of evapotranspiration (expressed as RMSE(OL) - RMSE (DA)) in units of $W/m^2$, using ALEXI data as the reference. The bottom panel shows the changes in R of modeled GPP using solar induced fluorescence data from GOME-2 as the reference, expressed as R (DA) - R (OL). The warm colors represent improvements from DA and cool colors represent degradations resulting from DA.

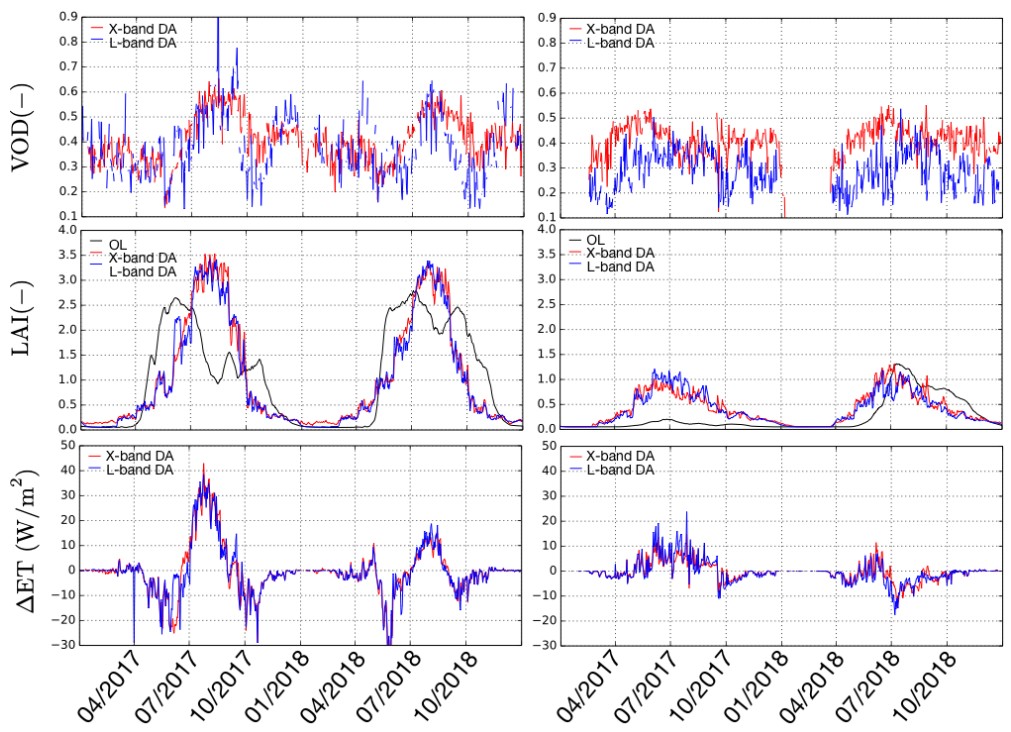

**Figure 7.** Time series of VOD (top panel), LAI (middle panel), and changes in evapotranspiration relative to the OL (bottom panel), for years 2017 and 2018, averaged over a cropland (location A in Figure 1) and woodlands (location B in Figure 1) area. The left and the right columns represent locations A and B, respectively.

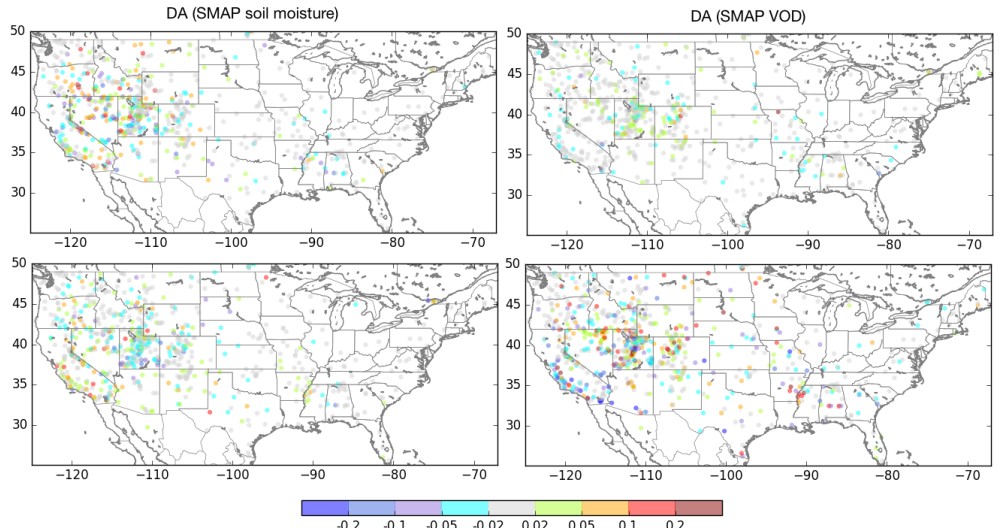

**Figure 8.** Differences in anomaly R values for surface soil moisture (top row) and root zone soil moisture (bottom row) from the assimilation of soil moisture (left column) and VOD (right column), relative to the OL integration. The warm and cool colors indicate improvements and degradations from DA. The gray shading indicates locations where the anomaly R differences are not statistically significant.

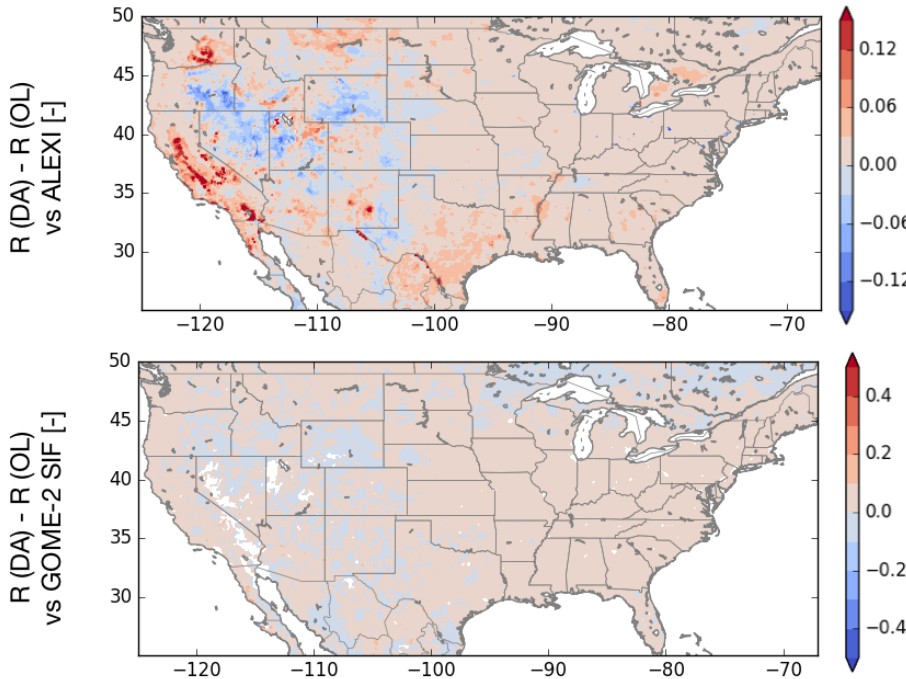

**Figure 9.** Differences in R values for ET (top row) and GPP (bottom row) from the assimilation of and VOD
relative to the OL integration, using ALEXI ET, and GOME-2 SIF datasets. The warm and cool colors indicate
improvements and degradations from DA.



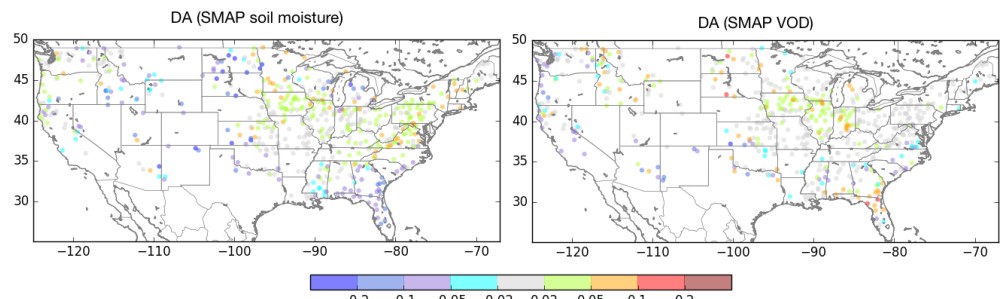

**Figure 10.** Improvements in streamflow NSE shown as NIC using the USGS daily streamflow observations as the reference.



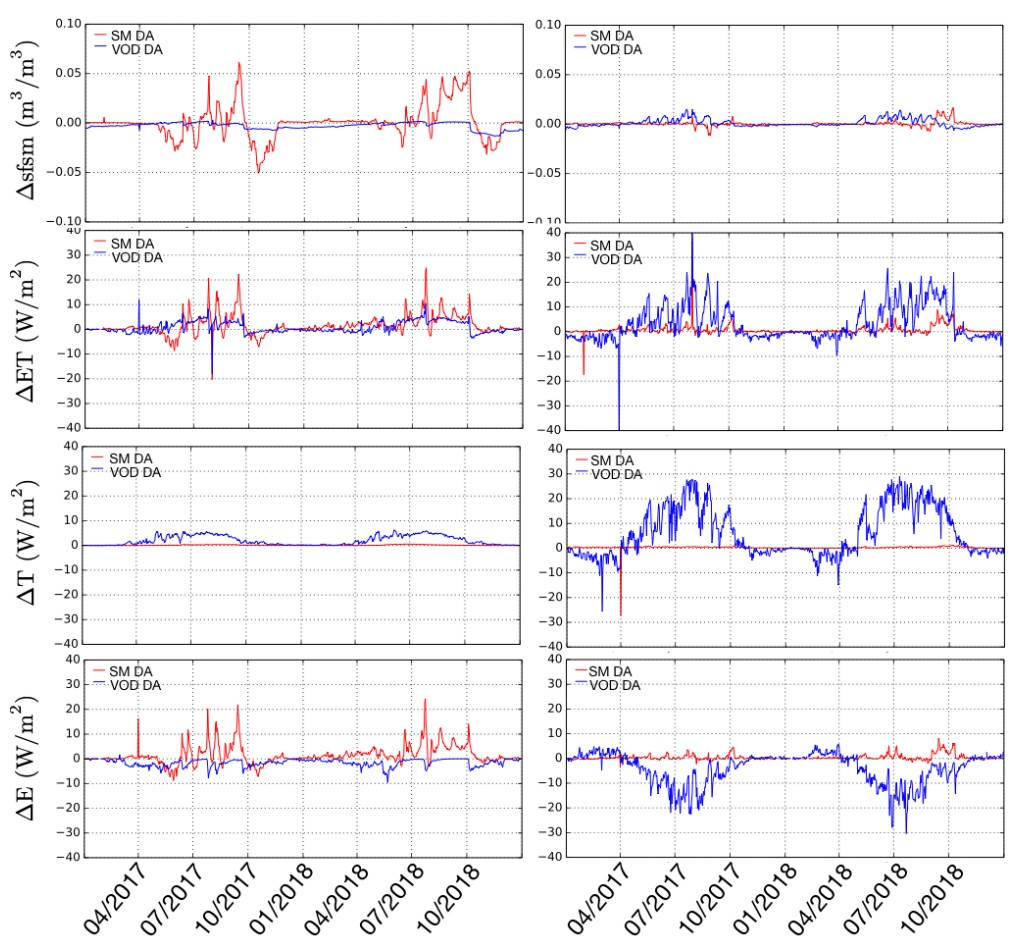

**Figure 11.** Time series of changes (relative to the OL) in ET, surface soil moisture, transpiration and bare soil evaporation for years 2017 and 2018, at locations C (western U.S.) and D (eastern U.S.). The left and the right columns represent locations C and D, respectively.



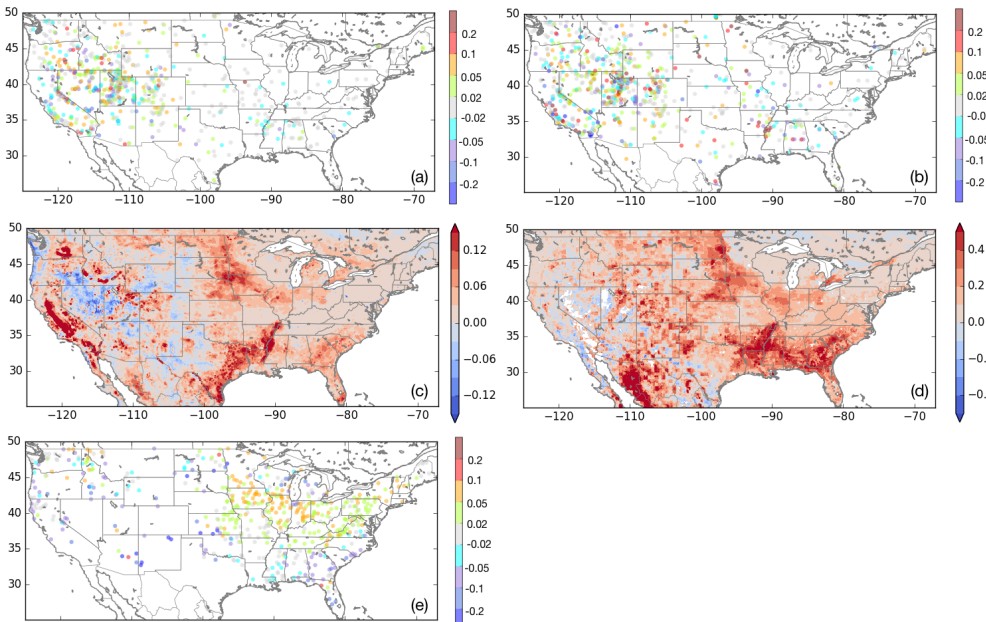

**Figure 12.** Impact of jointly assimilating SMAP surface soil moisture and VOD retrievals on surface soil moisture (a), root zone soil moisture (b), ET (c), GPP (d), and streamflow (e). Panels a and b show differences in anomaly R values using ISMN data as the reference; Panel c shows the differences in R values for ET using ALEXI as the reference dataset; Panel d shows the difference in R values for GPP with GOME-2 SIF retrievals as the reference; Panel e shows the NIC in streamflow using USGS daily streamflow observations as the reference. In each panel, the differences in the metric of evaluation are computed relative to the OL.



**Table 1.** Comparison of the percentage improvements in domain averaged skill metrics DA configuration that assimilates MODIS LAI (from Kumar et al. (2019b)), and the DA configurations that employ X- and C-band VOD retrievals, for different variables. SFSM - surface soil moisture, RZSM-root zone soil moisture, ET - evapotranspiration, GPP - gross primary productivity, TWS - terrestrial water storage, SF - streamflow

| Variable | Reference data | Metric (units) | DA (LAI) | DA-VOD (X-band) | DA-VOD (C-band) |
|----------|---------------|----------------|----------|-----------------|-----------------|
| SFSM | ISMN | Anomaly R (-) | 0.0 | 0.7 | 0.6 |
| RZSM | ISMN | Anomaly R (-) | 0.0 | 2.6 | 1.5 |
| ET | FLUXNET MTE | RMSE (W/m2) | 6.5 | 6.8 | 4.6 |
|  | ALEXI |  | 3.3/1.9 | 4.0/2.7 | 3.1/2.0 |
| GPP | FLUXCOM | RMSE (g/m2s) | 21.8 | 22.3 | 17.3 |
|  | GOME-2 SIF | R (-) | 17.0 | 24.5 | 17.0 |
| TWS | GRACE | Anomaly R (-) | 6.0 | 6.6 | 6.8 |
| SF | USGS | RMSE (m3/s) | 1.3 | 1.8 | 1.4 |