# Peer review of "Assimilation of vegetation optical depth retrievals from passive microwave radiometry"

_Hydrology and Earth System Sciences, 2020_

## Referee Comment (RC1) · Anonymous Referee #1 · 30 Mar 2020

Assimilation of vegetation optical depth retrievals from passive microwave radiometry Kumar et al., 2020

This manuscript shows the impact of microwave-based VOD and/or soil moisture data assimilation into the Noah-MP as part of LIS. The results are extensively evaluated using a wide set of independent estimates of various variables (incl. evapotranspiration, GPP, soil moisture, discharge). Overall, this is a great paper, worthy of publication after some clarifications and corrections.

Methodology:

- Why is VOD rescaled to MODIS LAI (GLASS) instead of rescaling it to the model LAI? There may be a large bias between the MODIS LAI and model LAI, which would violate

the Kalman filter assumptions. Perhaps show the spatial map of RMSD between the model LAI and the VOD after transformation to LAI (via GLASS)?

Another main concern with GLASS is that this product is filled with climatological values. Optical data do not have the same good coverage as microwave data (see also comment below). By now mapping VOD to GLASS, we basically undermine a key advantage of microwave data, i.e. we destroy the VOD information by mapping it to climatological LAI where insufficient LAI data are available...

- Why is VOD rescaled instead of installing an observation operator (H) that maps the model LAI to VOD? The latter would have the advantage that the Kalman gain would be able to capture more of the dynamic errors.

- Why is VOD (after rescaling) not bias-corrected, whereas soil moisture is?

- Isn't the SMAP VOD simply pre-calculated before retrieving soil moisture? The ATBD says that SMAP VOD is based on optical data (NDVI & stem index) and then used as an ancillary input to the soil moisture retrieval. It is then not surprising at all that the SMAP VOD corresponds more to optical LAI estimates (L. 222).

- Regardless of how SMAP VOD is pre-calculated or retrieved, the SMAP VOD and soil moisture estimates will have strongly correlated observation errors. Are these accounted for? If not, at least the individual errors should be increased to compensate for this lack or error correlations.

- The microwave retrievals are not at the same resolution as the model 1/8ˆo resolution. How does the 1-dimensional filter then work? There has to be some down- or upscaling.

- Data assimilation update vector: can you explicitly state the content of the update vector and does it change between VOD and SM assimilation experiments? (I do not think the vector should change with the experiment, but in between the lines of the text, I had the impression that it was changed; if done right, the update will naturally go

where it needs to go).

- Are the perturbations for all DA and OL experiments exactly the same?

- Soil moisture is rescaled via CDF-matching on a monthly basis. Is this monthly using multi-year information, or year by year?

Results:

- Fig 5: how is the change in unbiased RMSE or anomaly correlation for ET and GPP?

- L. 345: monthly mean? Year by year or multi-year means?

- L. 428: "seasonality in the anomalies and not the mean signal is the key factor in the CDF-matching"? But the CDF matching exactly tries to harmonize the mean signal of the observations and simulations. Rephrase?

- Around L. 510: one of the key results of the paper is in this paragraph and only supported by 2 time series at single points. It would be nice to have a more robust or convincing figure. For example, the correlation between RMSD(DA-OL) vs long-term mean soil moisture and vegetation for various DA experiments for all pixels, or something else that is spatially covering?

Textual issues:

- L. 70: typo guaranteed

- L. 177: write "1d"

- L. 292: pattern (without s; verb is singular)

- L. 340: indicate earlier on that the results are not shown.

- L. 365-368, Table 1: text and caption are cumbersome, consider rewriting to be more precise. Table 1 and caption are not clear. Caption first line "*for* DA configuration*s*"? These are percentage improvements *relative to the OL*? What are the 2 numbers in the evaluation against ALEXI? What is the purpose of the units here? The values are

all percentages, no?

- (!) Fig 6: panels or caption are not correct (RMSE-R).

- (!) Fig 9: caption is not correct.

- L. 498: switch the sentence to start with location C and then location D. Confusing now.

- Fig 11: LAI for location B: this is troublesome. The model LAI shows a clear interannual difference. With the DA, this interannual difference is removed. I am afraid that here, the VOD values are possibly rescaled to a multi-year average GLASS climatology, which inherently would not hold any interannual variability.

- L. 568: capitalize Kalman

- L.577-583: do something else than starting with "Though" in 3 subsequent sentences.

---

## Referee Comment (RC2) · Anonymous Referee #2 · 31 Mar 2020

Several studies have indicated the great potential of VOD for characterising land surface dynamics. To my knowledge, this study is the first one to report on a large-scale assimilation of VOD retrieved from various satellite sensors into a land surface model with dynamic vegetation. Therefore, I recommend publishing it after addressing several concerns and clarifications.

My major issues:

The study refers to VOD as an estimate of above-ground biomass, which it is not. Although relationships between the two quantities exist, which depend also on the microwave frequency, it is not the same thing -> Rephrase throughout the manuscript.

To my knowledge, VOD (tau) retrievals from SMAP L2 are not independent of optical

observations but a function of (MODIS) NDVI. Thus, it is not allowed to correlate VOD with (MODIS) LAI or assess its assimilation against that of assimilating LAI. In principle, an indicator of vegetation productivity is assimilated.

The impact of the retrieval algorithm on the results is unclear. For a robust comparison of the perofrmance of the different frequencies, I strongly recommend using the same retrieval algorithm for all frequencies.

It is unclear which VOD data are exactly assimilated. VODCA provides merged C-, X, and Ku-band products based on multiple sensors. Apart from the AMSR sensors mentioned, VODCA C- and X-band products alo use TRMM TMI and Windsat observations.

Line 397ff: it is surprising that the assimilation of L-band VOD gives results similar to those of X-band VOD, particularly because, as mentioned earlier, L-band is less sensitive to vegetation. Is this because you are assimilating NDVI rather than VOD (see my comment above)? Also provide quantitative results in addition to pattern descriptions.

Section 3.5: soil moisture and VOD are both derived from SMAP, which makes them strongly dependent. Do your assimilation operator account for these covariances? I recommend using soil moisture from SMAP and VOD from one of the other frequencies instead. In addition, for comaprability, can you show difference maps of the univariate and multivariate assimilation?

Some smaller issues:

line 8, line 21: do you really mean vegetation indices (i.e. spectral band ratio like NDVI) or vegetation variables (e.g LAI, GPP, biomass etc.)?

correct water limited -> water-limited, energy limited -> energy-limited, etc. when used as adjective.

line 31; for vegetation monitoring 70-100 ,m resolutio is not considered high-resolution

line 37: Although the benefits of passive MW are clearly acknowledged, it is also has diadvantages in terms of temporal resolution. -> add to manuscript

line 62: why do passive MW observations provide the opportunity to extend the spatial and temporal coverage when solar-reflective observations have been available globally for almost 50 years?

line 70: guranteed (typo)

The work of Teubner et al., 2018, 2019 [1,2] should be acknowledged wrt the relationship VOD-GPP.

Line 143: reference to the SMAP mission and the product used in this study shall be given.

Line 210: reference to Vreugdenhil et al. [3,4], who developed the ASCAT VOD product shall be provided

Line 223: This is not surprising as the 6.9 GHz C-band channel in the eastern US is strongly affected by RFI, whereas with SMAP you indirectly assimilate MODIS NDVI.

Line 228: Why does the rescaling not work in the southwestern US?

Line 278: I recommend using the more recent FLUXCOM roduct (Tramontana, 2016)

Line 319: In terms of radiative transfer mechanisms this is a very strong generalisation. Canyou provide the statistics for each category separately?

Line 329: phrased a bit unclear -> rephrase

Line 340: In the terms and conditions of the ISMN (https://ismn.geo.tuwien.ac.at/en/terms-and-conditions/) it is stated that reference (incl. citations) shall be given to all networks used -> please add

Line 342: which depths were used?

Lines 348-363: Since these results are not shown, I suggest moving these analyses to

a supplement

Line 385: Most LAI products are also derived from LEO orbits

Line 421: Isn't this more a bias correction?

[1] Teubner, I.E., Forkel, M., Jung, M., Liu, Y.Y., Miralles, D.G., Parinussa, R., van der Schalie, R., Vreugdenhil, M., Schwalm, C.R., Tramontana, G., Camps-Valls, G., Dorigo, W., 2018. Assessing the relationship between microwave vegetation optical depth and gross primary production. International Journal of Applied Earth Observation and Geoinformation 65, 79–91. https://doi.org/10.1016/j.jag.2017.10.006

[2] Teubner, I.E., Forkel, M., Camps-Valls, G., Jung, M., Miralles, D.G., Tramontana, G., van der Schalie, R., Vreugdenhil, M., Mösinger, L., Dorigo, W.A., 2019. A carbon sink-driven approach to estimate gross primary production from microwave satellite observations. Remote Sensing of Environment 229, 100–113. https://doi.org/10.1016/j.rse.2019.04.022

[3] Vreugdenhil, M., Dorigo, W., Wagner, W., de Jeu, R., Hahn, S., van Marle, M., 2016. Analysing the vegetation parameterisation in the TU-Wien ASCAT Soil Moisture Retrieval. IEEE Transactions on Geoscience and Remote Sensing 54 (6), 3513-3531. doi: 10.1109/TGRS.2016.2519842

[4] Vreugdenhil, M., Hahn, S., Melzer, T., Bauer-Marschallinger, B., Reimer, C., Dorigo, W., Wagner, W., 2017. Characteristing vegetation dynamics over Australia with ASCAT. IEEE Journal of Selected Topics in Applied Earth Observations and Remote Sensing 10 (5), 2240-2248, doi: 10.1109/JSTARS.2016.2618838

---

## Referee Comment (RC3) · Anonymous Referee #3 · 3 Apr 2020

This paper presents results of VOD data assimilation in the Noah-MP land surface model and its impact on soil moisture GPP, ET and streamflow. VOD products from AMSR-2 at X-Band and C-band are used, and SMAP VOD is assimilated separately and jointly with SMAP soil moisture products. The topic is highly relevant for to land surface scientific community. The paper is very well written, results are clearly presented and validated against a large range of observation types, and the analysis of the results is very thorough. I suggest the paper to be published in HESS after the suggestions below are considered.

Specific comments:

Abstract, lines 11-13: "The results also indicate that the independent information on moisture and vegetation states from SMAP can be simultaneously exploited through

the joint assimilation of surface soil moisture and VOD.": I don't agree with "independent information" as moisture and vegetation states are from the same sensor. Also, this sentence repeats line 9 and do not provide additional information. I suggest removing this sentence.

Line 70: It would be worth mentioning the Copernicus CIMR candidate mission (http://www.cimr.eu/). It will include all these frequencies. Although its primary objectives are related to sea ice and SST, it will be very relevant for VOD.

Lines 96-97: ". NASA's Soil Moisture Active Passive (SMAP; Entekhabi et al. (2010)) mission operates in a protected L-band, which minimizes the impact of RFI contamination." You should specify " over the US" because, unfortunately, L-band is much affected by RFI sources in many other regions.

Page 7 top paragraph (lines 194-205): As clearly explained in this section, X-band, C-band and L-band VODs are CDF matched to the GLASS LAI data so that they can be assimilated. However, it is not clear why GLASS LAI CDFs are computed (last sentence). Please clarify.

Page 7: Is there any quality control applied to the VOD data set before assimilation?

Page 8, lines 261-262: are SMAP VOD and soil moisture correlated observation errors accounted? The authors should clarify, and comment and justify the choice made in this study.

Page 10, lines 333-334: I find it confusing to give domain improvements in RMSE (in addition to R) for the comparison against ALEXI in these two sentences. The figures only present R statistics against ALEXI as explained on the previous page.

Page 11, line 340: "The impact of VOD assimilation on other land surface states such as soil moisture, terrestrial water storage, and streamflow is also evaluated using a number of reference products." Soil moisture and TWS validation results for the OL and the VOD DA experiments are discussed but results of streamflow validation are not

given in this sub-section. There are streamflow validation results in the next subsection but not comparing VOD DA with the OL. So, it would be interesting in section 3.1 to give streamflow validation results for C-band and X-band VOD DA compared to the open-loop.

Pages 11, lines 371-371 and Table 1: LAI DA has no impact on soil moisture. In this paragraph, the authors should comment on why.

Page 14, lines 453-455, Figure 8: the authors claim that the figure shows an overall improvement of soil moisture in the Western US. However, the figure shows a patchy impact in the Western US, with dominating blueish colours, which are related to degradation. It is perhaps an artefact of the figure which need to be made clearer.

Page 14, line 450 indicates that Figures 8 to 10 show results of SMOS soil moisture and VOD DA. It should be clarified that they show results in the univariate configurations. Also, the caption of Figure 9 has typos (see technical corrections).

Page 15, line510-515, and abstract line 11: the results presented in this paper clearly support the conclusion that soil moisture assimilation has more impact over water-limited areas. They also show that VOD assimilation has more impact in the eastern US and time series at location D shown in Figure 11 illustrate the impact very well. However, it is not convincing to conclude that VOD has an impact in energy-limited areas as patterns shown in eastern US and point D are not particularly energy limited, with point D is at latitude ~33 degrees North. The way it is formulated in the general conclusion line 597-599 is more correct (beneficial in areas with high vegetation and no water limitation). So, the abstract and the discussion page 15 should be updated accordingly.

Technical corrections

Line 110: 'independent reference datasets' is too vague. Please clarify.

Line 327: (Reichle and Koster ( 2004))

Line 328: over bare soil and urban areas

Line 332: "4.6 % and 6.8 %"

Figure 9 caption: 'of and VOD' -> 'of SMAP soil moisture and VOD'

―――――――――――――――――――――

---

## Author Comment (AC1) · 30 Apr 2020

**Referee 1**

Assimilation of vegetation optical depth retrievals from passive microwave radiometry
Kumar et al., 2020

This manuscript shows the impact of microwave-based VOD and/or soil moisture data assimilation into the Noah-MP as part of LIS. The results are extensively evaluated using a wide set of independent estimates of various variables (incl. evapotranspiration, GPP, soil moisture, discharge). Overall, this is a great paper, worthy of publication after some clarifications and corrections.

We really appreciate your thoughtful comments and have made significant changes to the manuscript in response to your suggestions. Please see below for our responses to your specific questions.

Methodology:

- Why is VOD rescaled to MODIS LAI (GLASS) instead of rescaling it to the model LAI? There may be a large bias between the MODIS LAI and model LAI, which would violate the Kalman filter assumptions. Perhaps show the spatial map of RMSD between the model LAI and the VOD after transformation to LAI (via GLASS)?

Thanks for the comment. The reviewer is correct if there are large and systematic biases between the model LAI and the MODIS LAI, the rescaling approach used here would be problematic. The approach was employed here based on the findings from the previous study assimilating GLASS LAI (Kumar et al., JHM 2019), which showed that the overall bias between the model and the observation was small (Figure 2 of that paper). Further, Kumar et al. 2019 study showed that the major improvements from assimilation are primarily from the correction of seasonality of vegetation, rather than from bias improvements (See Figures 1, 5, 6, and 7 in particular). Similar impacts are seen with the results in the current manuscript. For example, the time series at location A (Figure 7) shows the phase shift in ET introduced by VOD-DA, which results in an improvement in ET and GPP (Figure 3,4). The transformation of VOD into the LAI space, therefore, provides a quick way to enable the assimilation of VOD. To acknowledge this issue further, we have modified the description as follows on page 7:

"Note that the rescaling strategy used here also relies on the fact that the systematic errors between the GLASS LAI data and the NoahMP LAI are small, as demonstrated in Kumar et al. (2019b). In this prior study when GLASS LAI retrievals were assimilated within NoahMP, the demonstrated improvements were primarily from the adjustment of vegetation/crop seasonality, rather than from the correction of systematic errors. In addition, the positive impacts from the use of this strategy shown in the following sections, further confirm that this rescaling approach is reasonable."

Another main concern with GLASS is that this product is filled with climatological values. Optical data do not have the same good coverage as microwave data (see also

comment below). By now mapping VOD to GLASS, we basically undermine a key advantage of microwave data, i.e. we destroy the VOD information by mapping it to climatological LAI where insufficient LAI data are available.

The spatial gap-filling in the GLASS product is enabled by a general regression neural network (Xiao et al. 2014) (and not climatology) and prior studies have shown that the improved spatiotemporal coverage of the GLASS product has greater utility over that of the standard MODIS LAI product (Liang et al. 2014). The validation of the GLASS data and comparison against LAI products have also demonstrated the high quality of this product (Liao et al. (2012), Fang et al. (2013), Xiao et al. (2016)). Further, the previous study Kumar et al. (2019) demonstrated that the assimilation of GLASS LAI lead to significant improvements in the simulation of vegetation seasonality, water and carbon budget terms. The improvements in the simulation of vegetation seasonality over human managed agricultural areas were an important outcome of this study. The results in Kumar et al. (2019) confirm that the improvements from assimilation are not simply because of climatological improvements. The time series comparisons in this paper show that the variability in the VOD time series is preserved even after rescaling (Figure 7).

- Why is VOD rescaled instead of installing an observation operator (H) that maps the model LAI to VOD? The latter would have the advantage that the Kalman gain would be able to capture more of the dynamic errors.

Thank you for raising this point. The use of an observation operator (forward model) that maps LAI to VOD is another possible approach to assimilating VOD and we agree that it would enable capturing the dynamic errors. Note that we have already acknowledged this as a natural extension of this study on page 19 as:

"As noted in the description of the data assimilation methodology, the VOD retrievals are assimilated by rescaling them to the GLASS MODIS LAI climatology. This approach was employed as the prior study Kumar et al. (2019b) demonstrated significant positive impacts from the assimilation of the GLASS LAI data. Such an approach is needed also because the LSM does not have a prognostic representation of VOD. Though the beneficial impacts observed in the results indicate that this is a reasonable strategy, the rescaling essentially ignores the information on vertical heterogeneity in the canopy from these sensors. For example, the X-band data is documented to be more sensitive to the vegetation, whereas the L-band data is more representative of the lower canopy. A more direct use of the VOD data is likely to help in resolving these sensitivities within modeling. Extensions to this study that either uses a prognostic representation of VOD or a forward model that simulates VOD will enable such approaches. The current study serves as a useful benchmark for such future efforts."

- Why is VOD (after rescaling) not bias-corrected, whereas soil moisture is?

There are a number of reasons for employing bias-correction (and essentially assimilating anomalies only) for soil moisture DA, whereas the VOD is assimilated directly after rescaling to LAI. Direct assimilation of soil moisture retrievals is difficult because there are significant differences between the model estimates and satellite retrievals, in terms of their geophysical definitions and horizontal and vertical representativeness. Since the modeled soil moisture is essentially and index of wetness, a highly model-dependent quantity (Koster et al. 2009, Journal of Climate), it is generally inconsistent with satellite soil moisture retrievals and cannot be directly assimilated. As noted in our response about the rescaling of VOD, we use this approach based on success of directly assimilating LAI, as reported in Kumar et al. (2019). The text on page 8 clarifies these issues:

"Soil moisture in the LSMs is a model-specific quantify, an index of the moisture state (Koster et al. (2009)). As a result, there are significant differences in soil moisture estimates from different LSMs, even when forced with the same meteorology and land surface parameters (Dirmeyer et al. (2006)). Similarly, remote sensing based estimates of soil moisture are also indirect measurements generated through a retrieval model from direct measurements of the microwave emission of the land surface. Therefore, direct assimilation of soil moisture without resolving these inconsistencies is meaningless. Here we apply the commonly used strategy of CDF-matching (Reichle and Koster (2004)) to address the relative differences between the remote sensing and LSM-based soil moisture by rescaling the soil moisture retrievals into the LSM climatology before assimilation."

- Isn't the SMAP VOD simply pre-calculated before retrieving soil moisture? The ATBD says that SMAP VOD is based on optical data (NDVI & stem index) and then used as an ancillary input to the soil moisture retrieval. It is then not surprising at all that the SMAP VOD corresponds more to optical LAI estimates (L. 222).

The SMAP VOD used in this study is not the pre-flight VOD discussed in the ATBD. The VOD product used here is the SPL2SMP_E and its retrievals based on using both polarizations (V and H pol) to estimate soil moisture and VOD. We modified the description in section 2.1 as:

"The SMAP satellite launched in January 2015 is a mission dedicated to measuring soil moisture and freeze/thaw states, employing a passive microwave radiometer to collect measurements of vertical and horizontal polarizations of L-band brightness temperature data at an incident angle of $40°$. The retrievals from SMAP are also developed using the τ-ω model. The soil moisture retrievals are made using a single channel algorithm using the vertical polarizations (Chan et al. (2018)) whereas the VOD retrievals employ both polarized brightness temperature observations (Chaubell et al. (2020)). Though the sampling resolution of the SMAP radiometer is approximately 36 km, 150 oversampling of the antenna overpasses is used to enhance the spatial resolution to 9 km. This 9km, level 2 SMAP dataset (SPL2SMP–E) is used in this study. "

- Regardless of how SMAP VOD is pre-calculated or retrieved, the SMAP VOD and soil moisture estimates will have strongly correlated observation errors. Are these accounted for? If not, at least the individual errors should be increased to compensate for this lack or error correlations.

When SMAP VOD and soil moisture estimates are assimilated jointly, note that we simply combine two separate sequential assimilation instances (the observation vector does not consist of both VOD and soil moisture). In addition, the state vector used in these sequential assimilation instances are different. The soil moisture assimilation employs model soil moisture states whereas LAI is updated in the VOD assimilation instances. We have clarified this detail in the manuscript on page 17 as:

"As the results in the previous section indicate that assimilation of soil moisture and VOD can provide mutually exclusive information, an assimilation configuration that employs these retrievals simultaneously is developed. Note that in this joint configuration, rather than augmenting the observation vector to encompass both VOD and soil moisture retrievals, we simply combine the two separate sequential univariate assimilation instances within a single integration. Similar to the univariate configurations, in this multivariate configuration, soil moisture retrievals are used to update the surface soil moisture state, whereas VOD retrievals are used to update the prognostic LAI variable within the LSM."

- The microwave retrievals are not at the same resolution as the model 1/8ˆo resolution. How does the 1-dimensional filter then work? There has to be some down- or upscaling.

The 1d filter employs interpolated observations (using nearest neighbor approach) within the assimilation. This detail has been clarified in the text on page 7 as:

"In this study, the innovation calculations employ observations interpolated to the model grid using a nearest neighbor approach."

- Data assimilation update vector: can you explicitly state the content of the update vector and does it change between VOD and SM assimilation experiments? (I do not think the vector should change with the experiment, but in between the lines of the text, I had the impression that it was changed; if done right, the update will naturally go where it needs to go).

The state vector is not the same for VOD and SM experiments. In the VOD experiment, we update LAI and leaf biomass whereas in the SM experiment, we update the top soil moisture layer. These details are specified in the text on page 9 as:

"For VOD DA, additive perturbations with a standard deviation of 0.01 are applied to the model LAI fields (Kumar et al. (2019b)), every 3 hours. The updated LAI from DA is divided by the specific leaf area to revise the leaf biomass variable within Noah-MP. The state vector used in the soil moisture DA consists of the top soil moisture layer of Noah-MP, which is perturbed with an additive noise of 0.02 $m^3/m^3$, applied every 3 hours.

The perturbations also include time series correlations employed through a first order autoregressive (AR(1)) model with timescales of 24 and 3 hours, for the forcing and model state variables, respectively."

- Are the perturbations for all DA and OL experiments exactly the same?

There are no perturbations applied to the OL. In the DA experiments, the same exact perturbations are applied to the forcing variables. Similarly, the same set of model state perturbations are applied in all VOD DA configurations. Since the state vector used in the soil moisture DA is different, the state perturbations differ (as explained on page 9).

- Soil moisture is rescaled via CDF-matching on a monthly basis. Is this monthly using multi-year information, or year by year?

This is done monthly using a multi-year information.  We have added this clarification on page 7 as:

"Monthly CDFs using multi-year information are computed for both the VOD and LAI datasets using all available data, at every model grid point."

Results:
- Fig 5: how is the change in unbiased RMSE or anomaly correlation for ET and GPP?

Based on your query, we computed the changes in the unbiased RMSE and anomaly R for ET (using ALEXI as the reference) and GPP (using FLUXCOM as the reference). The figure below shows the % improvements from X-band and C-band DA. The results are similar to Figure 5, except that the level of improvements are smaller for anomaly oriented metrics. Improvements are also larger for moderate vegetation, similar to Figure 5. In the interest of not overwhelming the reader with additional metrics, we have not included these comparisons in the article.

[Figure]

– L. 345: monthly mean? Year by year or multi-year means?

The anomaly R values are computed based on multi-year monthly means. We have updated the text on page 11 to say:

"The anomaly R value at each grid point is computed based on daily soil moisture anomalies (of model and in-situ observations) calculated by subtracting the multi-year monthly mean values from the daily averages."

- L. 428: "seasonality in the anomalies and not the mean signal is the key factor in the CDF-matching"? But the CDF matching exactly tries to harmonize the mean signal of the observations and simulations. Rephrase?

We have rephrased this statement on page 14 as:

"Compared to location A, over the grassland location B, there are small climatological differences in the VOD retrievals from X- and L-band. These amplitudinal differences are reduced by the CDF matching, as the rescaled X- and L-band VOD estimates are similar to each other."

- Around L. 510: one of the key results of the paper is in this paragraph and only supported by 2 time series at single points. It would be nice to have a more robust or convincing figure. For example, the correlation between RMSD(DA-OL) vs long- term mean soil moisture and vegetation for various DA experiments for all pixels, or something else that is spatially covering?

Thanks for the comment. We respectfully disagree that contrasting the relative impacts from soil moisture and VOD DA are only supported by 2 time series at single points. Figure 8 essentially presents a spatially distributed comparison of long-term soil moisture, similar to the suggestion of the reviewer. Direct evaluations of soil moisture and vegetation is difficult as those sources are assimilated in our integrations. Figure 11 is supposed to supplement the spatially distributed evaluations of Figures 8-10, by drawing the contrast on the influence on the ET components. Reference datasets of the ET components (much less spatially distributed) are difficult to obtain. Therefore, we believe the current set of evaluations are sufficient to convey the key findings of the paper.

Textual issues:
- L. 70: typo guaranteed

Corrected
- L. 177: write "1d"

Corrected
- L. 292: pattern (without s; verb is singular)

Corrected
- L. 340: indicate earlier on that the results are not shown.

Though figures are not shown, the results of these evaluations are summarized in the text. We believe it is appropriate to provide the caveats of Figures not being shown for each comparison.

- L. 365-368, Table 1: text and caption are cumbersome, consider rewriting to be more precise. Table 1 and caption are not clear. Caption first line "*for* DA configuration*s*"? These are percentage improvements *relative to the OL*? What are the 2 numbers in the evaluation against ALEXI? What is the purpose of the units here? The values are all percentages, no?

The caption now reads:

"Comparison of the percentage improvements in domain averaged skill metrics (relative to the model OL) for DA configurations that assimilates MODIS LAI (from Kumar et al. (2019b)), and those that employ X- and C-band VOD retrievals, for different variables. "

The table has also been updated after removing the units. The evaluation against ALEXI shows the percentage improvements in RMSE and R. The table row has been updated to reflect this detail.

- (!) Fig 6: panels or caption are not correct (RMSE-R).

Corrected

- (!) Fig 9: caption is not correct.

Corrected

- L. 498: switch the sentence to start with location C and then location D. Confusing now.

Assuming that the reviewer is talking about line 488, the updated text on page 18 now reads:

"Location C is in the arid western U.S. with moderate vegetation, whereas location D is in the eastern U.S., representing a wet region with thick vegetation."

- Fig 11: LAI for location B: this is troublesome. The model LAI shows a clear interannual difference. With the DA, this interannual difference is removed. I am afraid that here, the VOD values are possibly rescaled to a multi-year average GLASS climatology, which inherently would not hold any interannual variability.

We assume that the reviewer is talking about Figure 7 instead of 11. In the figure, the reviewer is correct that the OL LAI shows an interannual difference unlike the DA time series. However, this is not due to the influence of GLASS climatology. Note that the rescaling of VOD is done to the GLASS data to generate LAI inputs for DA. Therefore, the model OL has no influence in the rescaling process. The time series of VOD (shown in the top panel) does not show a big interannual difference, which is why the rescaling does not show large interannual differences in LAI.

- L. 568: capitalize Kalman

Corrected

- L.577-583: do something else than starting with "Though" in 3 subsequent sentences.

The sentence on page 18 has been changed to say:

"The impacts on soil moisture, terrestrial water storage, and streamflow from VOD DA are found to be marginal."

---

## Author Comment (AC2) · 30 Apr 2020

Referee #2

Several studies have indicated the great potential of VOD for characterising land surface dynamics. To my knowledge, this study is the first one to report on a large-scale assimilation of VOD retrieved from various satellite sensors into a land surface model with dynamic vegetation. Therefore, I recommend publishing it after addressing several concerns and clarifications.

Thank you for the constructive comments. We have made significant changes to the manuscript to address your concerns. Please see below for our specific responses.

My major issues:

The study refers to VOD as an estimate of above-ground biomass, which it is not. Although relationships between the two quantities exist, which depend also on the microwave frequency, it is not the same thing -> Rephrase throughout the manuscript.

We have corrected these references to say that VOD is an 'analog' of above-ground biomass, instead of an 'estimate' of biomass.

To my knowledge, VOD (tau) retrievals from SMAP L2 are not independent of optical observations but a function of (MODIS) NDVI. Thus, it is not allowed to correlate VOD with (MODIS) LAI or assess its assimilation against that of assimilating LAI. In principle, an indicator of vegetation productivity is assimilated.

The SMAP L2 VOD observations are, in fact, independent of optical (MODIS) NDVI/LAI observations. We have explicitly stated with appropriate references in section 2.1 as:

"The SMAP satellite launched in January 2015 is a mission dedicated to measuring soil moisture and freeze/thaw states, employing a passive microwave radiometer to collect measurements of vertical and horizontal polarizations of L-band brightness temperature data at an incident angle of 40°. The retrievals from SMAP are also developed using the τ-ω model. The soil moisture retrievals are made using a single channel algorithm using the vertical polarizations (Chan et al. (2018)) whereas the VOD retrievals employ both polarized brightness temperature observations (Chaubell et al. (2020)). Though the sampling resolution of the SMAP radiometer is approximately 36 km, 150 oversampling of the antenna overpasses is used to enhance the spatial resolution to 9 km. This 9km, level 2 SMAP dataset (SPL2SMP−E) is used in this study. "

The impact of the retrieval algorithm on the results is unclear. For a robust comparison of the performance of the different frequencies, I strongly recommend using the same retrieval algorithm for all frequencies.

We agree that the study is not structured to compare and contrast the retrieval algorithm performance, as we simply use the available products. Developing VOD retrieval

products with the same algorithm and comparing them within a data assimilation framework is beyond the scope of this study. We have added the following caveat to acknowledge this limitation in the Summary section (pages 18 and 19).

"The study is conducted in the NLDAS-2 configuration over the Continental U.S. A suite of publicly available VOD retrievals from X-, C- and L-band instruments is assimilated in Noah-MP using a 1d ensemble Kalman filter algorithm. The X-and C-and retrievals from the Land Parameter Retrieval Model, whereas the L-band retrievals of VOD are from SMAP."

"It must be stressed that as the retrieval algorithms used to develop these VOD products are different, this particular study is not structured to assess the relative merits of each algorithm."

It is unclear which VOD data are exactly assimilated. VODCA provides merged C-, X, and Ku-band products based on multiple sensors. Apart from the AMSR sensors mentioned, VODCA C- and X-band products alo use TRMM TMI and Windsat observations.

The reviewer is correct that VOCA also uses data from WindSat, TMI, and GMI. We have updated the text as follows:

The text on pages 3 and 4 reads:

"As described in detail in Konings et al. (2017), a number of approaches have been used to retrieve VOD from microwave sensors. Here we employ VOD retrievals primarily from two approaches for data assimilation. The Land Parameter Retrieval Model (LPRM; Owe et al. (2008)) uses single frequency, polarized brightness temperature in the range of 1-20 GHz to retrieve both soil moisture and VOD. In this study, we use the C-band (6.9 GHz) and X-band (10.7 GHz) based VOD retrievals from LPRM. The C- and X-band measurements are less sensitive to cloud water content and more sensitive to soil moisture and vegetation canopy, which are also prone to Radio Frequency Interference (RFI). NASA's SMAP mission operates in a protected L-band over the U.S., which minimizes the impact of RFI contamination. The sensitivity of L-band to cloud water content is lower compared to C- and X-band. In addition, the L-band measurements provide more sensitivity to deeper soil moisture and canopy layers."

Pages 3 and 4 include the following description:

"In this study, we employ the VOD retrievals from LPRM version 6 (Van der Schalie et al. (2018)), available from the VOD climate archive (VODCA; Moesinger et al. (2019)). VODCA provides products from multiple sensors, including the Advanced Microwave Scanning Radiometer - Earth observing system (AMSR-E) aboard NASA's Aqua satellite, the AMSR2 instrument onboard the Global Change Observation Mission-Water (GCOM-W), WindSat microwave radiometer aboard the joint DoD/Navy Coriolis

platform, the Tropical Rainfall Measuring Mission's (TRMM) Microwave Imager (TMI) and the Global Precipitation Measurement (GPM) Microwave Imager (GMI). The C-band VOD retrievals rely on AMSR-E, AMSR2, and WindSat, whereas the X-band VOD retrievals include data from AMSR-E, AMSR2, WindSat, TMI, and GMI."

Line 397ff: it is surprising that the assimilation of L-band VOD gives results similar to those of X-band VOD, particularly because, as mentioned earlier, L-band is less sensitive to vegetation. Is this because you are assimilating NDVI rather than VOD (see my comment above)? Also provide quantitative results in addition to pattern descriptions.

As noted in the manuscript, the X-band and L-band evaluation results are presented for different time periods (X-band integrations cover a time period of 2000-2018 whereas L-band runs are only for 2015-2019). Since evaluation time periods also differ for X- and L-band VOD runs, the results for X- and L-band are not directly comparable. As noted in our earlier response, the assimilation of NDVI is not a factor in the SMAP VOD retrievals.

Section 3.5: soil moisture and VOD are both derived from SMAP, which makes them strongly dependent. Do your assimilation operator account for these covariances? I recommend using soil moisture from SMAP and VOD from one of the other frequencies instead. In addition, for comaprability, can you show difference maps of the univariate and multivariate assimilation?

When SMAP VOD and soil moisture estimates are assimilated jointly, we simply combine two separate sequential assimilation instances (the observation vector does not consist of both VOD and soil moisture, and therefore the explicit consideration of the covariances is not needed). In addition, the state vector used in these sequential assimilation instances are different. The soil moisture assimilation employs model soil moisture states whereas LAI is updated in the VOD assimilation instances. We also consider obtaining fresh retrievals from select frequencies to be outside the scope of this paper.

The manuscript has been updated with the following clarification on page 17:

"As the results in the previous section indicate that assimilation of soil moisture and VOD can provide mutually exclusive information, an assimilation configuration that employs these retrievals simultaneously is developed. Note that in this joint configuration, rather than augmenting the observation vector to encompass both VOD and soil moisture retrievals, we simply combine the two separate sequential univariate assimilation instances within a single integration. Similar to the univariate configurations, in this multivariate configuration, soil moisture retrievals are used to update the surface soil moisture state, whereas VOD retrievals are used to update the prognostic LAI variable within the LSM."

Some smaller issues:

line 8, line 21: do you really mean vegetation indices (i.e. spectral band ratio like NDVI) or vegetation variables (e.g LAI, GPP, biomass etc.)?

'Vegetation indices' is used to mean both estimates such as NDVI, LAI, GPP, and biomass.

correct water limited -> water-limited, energy limited -> energy-limited, etc. when used as adjective.

Thanks for the suggestion. All such instances have been corrected in the manuscript.

line 31; for vegetation monitoring 70-100 ,m resolution is not considered high-resolution

'high resolution' has been changed to 'fine resolution'

line 37: Although the benefits of passive MW are clearly acknowledged, it is also has disadvantages in terms of temporal resolution. -> add to manuscript

line 62: why do passive MW observations provide the opportunity to extend the spatial and temporal coverage when solar-reflective observations have been available globally for almost 50 years?

In the above two comments, we assume that the reviewer is referring to the diurnal coverage afforded by GEO-based optical/IR measurements in cloud free days. The following changes are made to the manuscript.

"Gap-filling strategies, such as using the nearest clear-day observation, are often used to improve the cloud- related gaps in spatio-temporal coverage from optical/TIR instruments (Hall et al. (2010))."

"As the use of all-weather VOD measurements from microwave sensors provides the opportunity to extend the spatial and temporal coverage of vegetation observations into overcast and clouded conditions, here we examine the influence of assimilating VOD retrievals from microwave radiometry."

line 70: guranteed (typo)

Corrected.

The work of Teubner et al., 2018, 2019 [1,2] should be acknowledged wrt the relation-ship VOD-GPP.

Thank you for suggesting these relevant references. They have been included in the discussion about prior studies examining VOD as an analog for vegetation conditions on page 2.

Line 143: reference to the SMAP mission and the product used in this study shall be given.

The reference to the SMAP mission (Entekhabi et al. 2010) is provided earlier in the text, at the first mention.

Line 210: reference to Vreugdenhil et al. [3,4], who developed the ASCAT VOD product shall be provided

Thanks for the reference. It has been included in the manuscript on page 8.

Line 223: This is not surprising as the 6.9 GHz C-band channel in the eastern US is strongly affected by RFI, whereas with SMAP you indirectly assimilate MODIS NDVI.

The SMAP VOD retrievals are not dependent on MODIS NDVI. The VOD retrievals are based on using both polarizations (V and H pol) to estimate soil moisture and VOD.

Line 228: Why does the rescaling not work in the southwestern US?

The correlation between LAI and VOD is weaker in the southwest US because the vegetation is sparse (Figure 1). As noted in the text the correlations are strong in areas with high vegetation density. The assimilation results also show a near neutral impact over the southwest from VOD assimilation.

Line 278: I recommend using the more recent FLUXCOM product (Tramontana, 2016)

Thank you for the suggestion. We computed the impact of X-band and C-band VOD DA using the FLUXCOM energy fluxes from 2001-2015. The improvements in RMSE from this comparison is shown below (X-band DA on the left, C-band DA on the right). The results are qualitatively similar to the FLUXNET MTE comparison shown in the paper. Given that the paper discusses comparisons to the optical sensor based LAI-DA results (which are already published with FLUXNET MTE data) and because the conclusions about the impact of DA remain unchanged, we have not included this updated comparison in the manuscript.

[Figure]

Line 319: In terms of radiative transfer mechanisms this is a very strong generalisation. Can you provide the statistics for each category separately?

The actual vegetation type used in the model simulations include 13 categories (Figure 1). As noted in the text, grouped categorization is used here for simplicity. The statistics for each category is provided in Figure 5.

Line 329: phrased a bit unclear -> rephrase

This statement on page 11 has been modified to:

"Over bare soil and urban areas, the impact of VOD assimilation is very small, due to the lack of vegetation influence on ET and GPP."

Line 340: In the terms and conditions of the ISMN (https://ismn.geo.tuwien.ac.at/en/terms-and-conditions/) it is stated that reference (incl. citations) shall be given to all networks used -> please add

Thanks for pointing out this detail. On page 11, we have updated the references to include additional as suggested on the 'terms and conditions' page (Note that the link to the "Networks" is not working at the moment, we included all the references that are available in the Readme.txt that comes with the data).

Line 342: which depths were used?

We used data up to 1 m of the root zone. This has been clarified in the text on page 11 as:

"The surface and root zone soil moisture values are  defined as the soil moisture content of the top 10 cm and 1 meter of the soil column, respectively. These are computed from the layer soil moisture values as suitably weighted vertical averages based on the thickness of the soil layers."

Lines 348-363: Since these results are not shown, I suggest moving these analyses to a supplement

Though additional figures are not included to describe these results, we think it is important to include them in the main manuscript, partly because the following sections (that contrast soil moisture and VOD DA) do include more detailed evaluations of these variables.

Line 385: Most LAI products are also derived from LEO orbits

We have rephrased the description on page 13 as:

"Note that the spatial resolution of passive microwave retrievals is typically coarser than those from the optical/IR sensors. In addition, passive microwave measurements are only available from low earth orbits (LEO) due to the antenna size requirements, so they can't provide the diurnal view as available for optical/IR instruments from geostationary satellites."

Line 421: Isn't this more a bias correction?

As noted in the text, the corrections are more to the phase of the vegetation seasonality from assimilation. Note that in the non-peak months, the changes to LAI from DA is small, which suggests that the main impact of assimilation is not a systematic bias correction. Yes, there are bias changes in the peak vegetation months, but those are important for fixing the seasonality, demonstrated in the evaluation of ET, GPP and other variables.

[1] Teubner, I.E., Forkel, M., Jung, M., Liu, Y.Y., Miralles, D.G., Parinussa, R., van der Schalie, R., Vreugdenhil, M., Schwalm, C.R., Tramontana, G., Camps-Valls, G., Dorigo, W., 2018. Assessing the relationship between microwave vegetation optical depth and gross primary production. International Journal of Applied Earth Observa- tion and Geoinformation 65, 79–91. https://doi.org/10.1016/j.jag.2017.10.006

[2] Teubner, I.E., Forkel, M., Camps-Valls, G., Jung, M., Miralles, D.G., Tra- montana, G., van der Schalie, R., Vreugdenhil, M., Mösinger, L., Dorigo, W.A., 2019. A carbon sink-driven approach to estimate gross primary production from microwave satellite observations. Remote Sensing of Environment 229, 100–113. https://doi.org/10.1016/j.rse.2019.04.022

[3] Vreugdenhil, M., Dorigo, W., Wagner, W., de Jeu, R., Hahn, S., van Marle, M., 2016. Analysing the vegetation parameterisation in the TU-Wien ASCAT Soil Moisture Retrieval. IEEE Transactions on Geoscience and Remote Sensing 54 (6), 3513-3531. doi: 10.1109/TGRS.2016.2519842

[4] Vreugdenhil, M., Hahn, S., Melzer, T., Bauer-Marschallinger, B., Reimer, C., Dorigo, W., Wagner, W., 2017. Characteristing vegetation dynamics over Australia with ASCAT. IEEE Journal of Selected Topics in Applied Earth Observations and Remote Sensing 10 (5), 2240-2248, doi: 10.1109/JSTARS.2016.2618838

---

## Author Comment (AC3) · 30 Apr 2020

Referee #3

This paper presents results of VOD data assimilation in the Noah-MP land surface model and its impact on soil moisture GPP, ET and streamflow. VOD products from AMSR-2 at X-Band and C-band are used, and SMAP VOD is assimilated separately and jointly with SMAP soil moisture products. The topic is highly relevant for to land surface scientific community. The paper is very well written, results are clearly presented and validated against a large range of observation types, and the analysis of the results is very thorough. I suggest the paper to be published in HESS after the suggestions below are considered.

Thanks you for the supportive and helpful comments. Please see below for our responses and the details of the changes made to the manuscript.

Specific comments:

Abstract, lines 11-13: "The results also indicate that the independent information on moisture and vegetation states from SMAP can be simultaneously exploited through the joint assimilation of surface soil moisture and VOD.": I don't agree with "independent information" as moisture and vegetation states are from the same sensor. Also, this sentence repeats line 9 and do not provide additional information. I suggest removing this sentence.

Thanks you for the comment. The 'independent' qualifier mainly applies over locations where vegetation is thick, since soil moisture is excluded from assimilation over those regions (whereas VOD is not). In any case, based on your suggestion, we have removed the 'independent' qualifier in the text, to avoid any confusion.

Line 70: It would be worth mentioning the Copernicus CIMR candidate mission (http://www.cimr.eu/). It will include all these frequencies. Although its primary objectives are related to sea ice and SST, it will be very relevant for VOD.

Thank you for the suggestion. We have added CIMR to the list of mentioned missions on page 3.

Lines 96-97: ". NASA's Soil Moisture Active Passive (SMAP; Entekhabi et al. (2010)) mission operates in a protected L-band, which minimizes the impact of RFI contamination." You should specify " over the US" because, unfortunately, L-band is much affected by RFI sources in many other regions.

We have added this additional clarification to the text on page 4.

Page 7 top paragraph (lines 194-205): As clearly explained in this section, X-band, C-band and L-band VODs are CDF matched to the GLASS LAI data so that they can be

assimilated. However, it is not clear why GLASS LAI CDFs are computed (last sentence). Please clarify.

Thanks for the comment. The GLASS LAI CDFs are employed here based on the findings from the previous study assimilating GLASS LAI (Kumar et al., JHM 2019). The transformation of VOD into the LAI space, provides a quick way to enable the assimilation of VOD. To acknowledge this issue further, we have modified the description on page 7 as follows:

"Note that the rescaling strategy used here also relies on the fact that the systematic errors between the GLASS LAI data and the NoahMP LAI are small, as demonstrated in Kumar et al. (2019b). In this prior study when GLASS LAI retrievals were assimilated within NoahMP, the demonstrated improvements were primarily from the adjustment of vegetation/crop seasonality, rather than from the correction of systematic errors. In addition, the positive impacts from the use of this strategy shown in the following sections, further confirm that this rescaling approach is reasonable."

Page 7: Is there any quality control applied to the VOD data set before assimilation?

Yes, a number of QC flags are applied to the VOD data. Retrievals are excluded near water bodies, for being at the edge of the swath and when soil is frozen/covered by snow. The description in the text on pages 8 and 9 has been modified to say :

"Similar to the strategy used in prior studies, soil moisture retrievals are excluded near water bodies, for being at the edge of the swath, when soil is frozen/covered by snow, and when the vegetation cover is thick (Kumar et al. (2019a)), to account for the known limitations of passive microwave-based soil moisture retrievals. Similar flags except for thick vegetation are also applied to screen out VOD retrievals."

Page 8, lines 261-262: are SMAP VOD and soil moisture correlated observation errors accounted? The authors should clarify, and comment and justify the choice made in this study.

When SMAP VOD and soil moisture estimates are assimilated jointly, we simply combine two separate sequential assimilation instances (the observation vector does not consist of both VOD and soil moisture, and therefore the explicit consideration of the covariances is not needed). In addition, the state vector used in these sequential assimilation instances are different. The soil moisture assimilation employs model soil moisture states whereas LAI is updated in the VOD assimilation instances. The manuscript has been updated with the following clarification on page 17:

"As the results in the previous section indicate that assimilation of soil moisture and VOD can provide mutually exclusive information, an assimilation configuration that employs these retrievals simultaneously is developed. Note that in this joint configuration, rather than augmenting the observation vector to encompass both VOD and soil moisture retrievals, we simply combine the two separate sequential univariate

assimilation instances within a single integration. Similar to the univariate configurations, in this multivariate configuration, soil moisture retrievals are used to update the surface soil moisture state, whereas VOD retrievals are used to update the prognostic LAI variable within the LSM."

Page 10, lines 333-334: I find it confusing to give domain improvements in RMSE (in addition to R) for the comparison against ALEXI in these two sentences. The figures only present R statistics against ALEXI as explained on the previous page.

The percentage improvements in RMSE are given in the text to allow the comparisons presented in Table 1 (since the LAI-DA results in Kumar et al. 2019 were only provided for RMSE).

Page 11, line 340: "The impact of VOD assimilation on other land surface states such as soil moisture, terrestrial water storage, and streamflow is also evaluated using a number of reference products." Soil moisture and TWS validation results for the OL and the VOD DA experiments are discussed but results of streamflow validation are not given in this sub-section. There are streamflow validation results in the next subsection but not comparing VOD DA with the OL. So, it would be interesting in section 3.1 to give streamflow validation results for C-band and X-band VOD DA compared to the open-loop.

Thank you for pointing out this omission. We have updated Section 3.1 with the following additional paragraph.

"The impact of VOD assimilation on streamflow is evaluated by comparing to the U.S. Geological Survey (USGS) daily gauge measurements at locations minimally impacted by reservoir operations (Kumar et al. (2014, 2019b)). The impact of DA is quantified using the Normalized Information Contribution (NIC) metric on Nash Sutcliffe Efficiency (NSE) of streamflow (Kumar et al. (2014)), with positive and negative NIC values indicating benefit and degradation from assimilation, respectively. Overall, there is a small, but beneficial impact from VOD assimilation on streamflow. The domain averaged NIC improvements from X-band and C-band VOD DA is 0.03 and 0.02, respectively, with larger improvements noticed over the agricultural areas of the Midwest U.S."

Pages 11, lines 371-371 and Table 1: LAI DA has no impact on soil moisture. In this paragraph, the authors should comment on why.

Thank you for pointing this out. Upon examining the % changes from the Kumar et al. (2019) LAI-DA study, we discovered that the 0% improvements reported for soil moisture were incorrect. The table has now been updated with the correct values, which report 0.6% and 2.3% improvements in surface and root zone soil moisture from LAI-DA.

Page 14, lines 453-455, Figure 8: the authors claim that the figure shows an overall improvement of soil moisture in the Western US. However, the figure shows a patchy impact in the Western US, with dominating blueish colours, which are related to degradation. It is perhaps an artefact of the figure which need to be made clearer.

The reviewer is correct that there is some patchiness (with degradations mixed in) in the western U.S. of Figure 8. However, the average changes in anomaly R west of 100W shows that the DA has a slightly more positive impact. For example, the domain averaged percentage improvements in surface and root zone from soil moisture DA over the western domain are 2.50% and 1.45%, respectively (compared to 2.14% and 1.30% for the whole domain). Similarly, for the assimilation of VOD, the domain averaged percentage improvements in surface and root zone soil moisture for the western domain are 0.28% and 0.7% (compared to 0.31% and 0.5%). Given that these domain improvements are small, we have changed the description on page 15 to acknowledge these facts, as:

"Figures 8 to 10 show the impacts of separately assimilating SMAP soil moisture and VOD retrievals on various land surface water and carbon states. Using the in-situ soil moisture measurements from ISMN as the reference, Figure 8 shows the changes in anomaly R of surface and root zone soil moisture from soil moisture and VOD assimilation. Overall, soil moisture DA has a positive impact on the simulation of surface soil moisture, particularly in the Western U.S. and Highplains. Approximately 2.1% improvement in domain averaged anomaly R is obtained from SMAP soil moisture assimilation. The impact of soil moisture DA over the Eastern U.S. is small, as these regions of high vegetation density are generally excluded from soil moisture DA. Comparatively, VOD assimilation has little impact on surface soil moisture, as the changes in anomaly R are not statistically significant in most locations. Both soil moisture and VOD assimilation also impact root zone soil moisture estimates, with varying levels of improvements and degradations across the domain. The assimilation of SMAP soil moisture improves the root zone estimates over the lower Mississippi and parts of the Western U.S. including California, Nevada, and Colorado. The patterns of improvements and degradations in root zone soil moisture are more mixed in the VOD assimilation results, over these same areas."

Page 14, line 450 indicates that Figures 8 to 10 show results of SMOS soil moisture and VOD DA. It should be clarified that they show results in the univariate configurations. Also, the caption of Figure 9 has typos (see technical corrections).

Thanks for the comment. We have modified the line on page 15 to say : "Figures 8 to 10 show the impacts of separately assimilating SMAP soil moisture and VOD retrievals on various land surface water and carbon states."

The caption of Figure 9 has been updated as well.

Page 15, line 510-515, and abstract line 11: the results presented in this paper clearly support the conclusion that soil moisture assimilation has more impact over waterlimited areas. They also show that VOD assimilation has more impact in the eastern US and time series at location D shown in Figure 11 illustrate the impact very well. However, it is not convincing to conclude that VOD has an impact in energy-limited areas as patterns shown in eastern US and point D are not particularly energy limited, with point D is at latitude ~33 degrees North. The way it is formulated in the general conclusion line 597-599 is more correct (beneficial in areas with high vegetation and no water limitation). So, the abstract and the discussion page 15 should be updated accordingly.

Thank you for the comment and we agree. The following changes are made to the manuscript:

Abstract now reads:

"The utility of soil moisture assimilation for improving ET is more significant over water-limited regions, whereas VOD DA is more impactful over areas where soil moisture is not the primary controlling factor on ET."

The corresponding discussion in the text on page 16 has been modified to say:

"Over areas with high vegetation and little water limitation, vegetation growth and stomatal control, more than surface moisture conditions, influence the ET evolution."

Technical corrections
Line 110: 'independent reference datasets' is too vague. Please clarify.

We have modified the text on page 4 to say:

"These questions are addressed by examining the impact of assimilation with the use of a large suite of independent reference datasets of soil moisture, evapotranspiration, gross primary productivity (GPP), streamflow and terrestrial water storage (TWS)."

Line 327: (Reichle and Koster ( 2004))

Corrected

Line 328: over bare soil and urban areas

Corrected
Line 332: "4.6 % and 6.8 %"

Corrected
Figure 9 caption: 'of and VOD' -> 'of SMAP soil moisture and VOD'

Corrected